



# Machine learning models to predict myocardial infarctions from past climatic and environmental conditions

Lennart Marien[1], Mahyar Valizadeh[2,*], Wolfgang zu Castell[3], Christine Nam[1], Diana Rechid[1], Alexandra Schneider[2], Christine Meisinger[4,5], Jakob Linseisen[4,5], Kathrin Wolf[2,**], and Laurens M Bouwer[1,*,**]

[1]Climate Service Center Germany (GERICS), Helmholtz-Zentrum Hereon, Fischertwiete 1, 20095, Hamburg, Germany
[2]Institute of Epidemiology, Helmholtz Zentrum München, German Research Center for Environmental Health, Ingolstädter Landstraße 1, 85764, Neuherberg, Germany
[3]GFZ German Research Centre for Geosciences, Telegrafenberg, 14473, Potsdam, Germany
[4]Chair of Epidemiology, University of Augsburg, University Hospital Augsburg, Stenglinstr. 2, 86156 Augsburg
[5]Clinical Epidemiology, Helmholtz Zentrum München, German Research Center for Environmental Health, Ingolstädter Landstr. 1, 85754 Neuherberg
[*]These authors contributed equally to this work.
[**]These authors share last authorship.

**Correspondence:** Laurens M. Bouwer (laurens.bouwer@hereon.de)

**Abstract.** Myocardial infarctions (MI) are a major cause of death worldwide, and temperature extremes, e.g., during heat waves and cold winters, may increase the risk of MI. The relationship between health impacts and climate is complex and is influenced by a multitude of climatic, environmental, socio-demographic, and behavioral factors. Here, we present a Machine Learning (ML) approach for predicting MI events based on multiple environmental and demographic variables. We derived data on MI

events from the KORA MI registry dataset for Augsburg, Germany between 1998 and 2015. Multivariable predictors include weather and climate, air pollution ($PM_{10}$, NO, $NO_2$, $SO_2$, and $O_3$), surrounding vegetation, as well as demographic data. We tested the following ML regression algorithms: Decision Tree, Random Forest, Multi-layer Perceptron, Gradient Boosting and Ridge Regression. The models are able to predict the total annual number of MI reasonably well (adjusted $R^2 = 0.59 - 0.71$). Inter-annual variations and long-term trends are captured. Across models the most important predictors are air pollution and

daily temperatures. Variables not related to environmental conditions, such as demographics need to be considered as well. This ML approach provides a promising basis to model future MI under changing environmental conditions, as projected by scenarios for climate and other environmental changes.

## 1  Introduction

Myocardial infarctions (MI), commonly known as heart attacks, are a major cause of death worldwide, responsible for a

substantial amount of cardiovascular related mortality, as well as a major contributor to disability. The estimated prevalence of MI worldwide in 2015 was close to 16 million, with 33.000 years lived with disability attributed to the condition (Vos et al., 2016). In light of ageing western societies as well as ongoing environmental and climatic change, MI is likely to remain





a considerable burden to health systems in the future. It is therefore paramount to deepen the understanding of the complex interplay between environmental and other risk factors and their effect on MI and to estimate their expected future development.

Epidemiological research has shown that air temperature extremes, cold as well as heat, can play an important role in triggering acute MI (Chen et al., 2019; Wolf et al., 2009). This is especially apparent in winter when most of the MI events are observed. This is not only linked to the fact that cold temperatures are a known risk factor for MI (e.g., The Eurowinter Group, 1997; Schwartz et al., 2004; Wolf et al., 2009; Bhaskaran et al., 2010) but possibly also to the coincidence of respiratory diseases during that time of the year exerting physical stress and increasing vulnerability as evidenced by increased clotting

activity and inflammatory processes (Woodhouse et al., 1994). At the same time, severe periods of heat as encountered during heat waves are likely to occur with higher frequency, intensity, and duration due to anthropogenic climate change even if limited to warming levels between 1.5° and 2° (Sieck et al., 2020). Increasing levels of urbanisation entail higher levels of exposure to heat as well as a consequence of the urban heat island effect (e.g., Feng et al., 2014; Zhang et al., 2009). Moreover, the elderly are particularly vulnerable to MI, exacerbating the potential adverse effects in light of the demographic ageing expected in

developed countries, such as Germany (Schmidt et al., 2013).

    A key issue in understanding current and future health impacts is the inclusion of a multitude of processes and circumstances that influence the health outcomes (Roth, 2020), such as MI, in quantitative models. These include environmental influences, such as the occurrence of high and low temperature events, air quality, the presence of water bodies and vegetation and characteristics of the built environment. Humidity is often included as another important meteorological variable in studying human

health impacts (Davis et al., 2016). For instance, high temperatures are often perceived as more straining under very humid conditions, such as in case of sultriness. Hot and strongly saturated air carries less oxygen and interferes with transpiration as one of the main mechanisms of cooling the human body (Havenith, 2005). Therefore, the same temperature can be perceived more strongly if humidity is high as well. But the relevance of humidity for myocardial infarctions has not been confirmed (e.g., Schwartz et al., 2004). Changes in the exposed population (age, sex), their health status and underlying diseases are im-

portant as well. Therefore future health risks from climate change cannot only be explained from changes in extreme weather, however many studies in fact do so for projecting future health risks under climate change and these have been criticised for this (Vanos et al., 2020). Finally, health interventions such as heat health action plans, and overall better healthcare have been shown to reduce health risks from extreme temperatures to a significant extent (see for instance Achebak et al., 2019). But also policies related to climate change, such as reduced traffic emissions, are expected to lead to better health outcomes (Laverty

et al., 2021).

    So, for more reliable estimations of potential future risks, multiple variables must be incorporated in a ML model. In addition, several of the relations between environmental factors and health outcomes are only partially known. This is where data-driven approaches are particularly useful, as they can provide accurate estimations of complex processes, taking up many variables and also account for complex and non-linear relations between those variables. Machine Learning (ML) approaches are now

being tested widely for earth system modelling and other environmental studies (Reichstein et al., 2019), and they are also increasingly used to estimate the social and economic impacts of environmental extremes such as floods and windstorms (Merz et al., 2013; Wagenaar et al., 2017, 2021). ML however, has only recently been applied to health impact modeling.



Several studies have employed statistical methods as well as ML to predict infectious diseases, such as malaria transmissions (Zinszer et al., 2012; Sewe et al., 2017). Zhang et al. (2014) studied heat-related mortality, and identified relevant temperature and humidity variables using Random Forests. Other studies applied ML to evaluate risk of MI or to predict acute MI based on data such as patient history, blood markers, or electrocardiogram, but lack an environmental dimension (e.g., Tamarappoo et al., 2021; Commandeur et al., 2020).

In this study we correlate the total number of MI events on a given day to the coinciding and preceding environmental hazards. We employ several machine learning algorithms in a data-driven setting, using a range of meteorological, environmental, demographic and health variables. Our hypothesis is that the ML algorithm will predict a higher number of MI events when temperatures are extremely high or extremely low and/or when other environmental conditions deteriorate, for instance when air pollution is high. We estimate the importance of the predictive variables in the models. We also assess the effects on different sub-groups, depending on location (urban/rural) as these may exhibit different vulnerabilities (Gabriel and Endlicher, 2011), and patient properties (age, smoking, and diabetes). The ML models that are presented can be used to estimate future health outcomes, using a set of scenarios for changes in climatic, environmental and demographic variables.

We expect however that none of the risk factors that are included in our models is strong enough to directly trigger MI in an otherwise healthy person. Instead, these environmental and demographic factors must be assumed to increase the statistical likelihood of vulnerability to MI over longer periods of time. Many common root causes of MI, e.g., atherosclerosis, are chronic and acquired over time and usually in large part due to lifestyle risks such as consumption of alcohol, tobacco, obesity, emotional stress, lack of exercise. Some of the risk factors covered in this study, e.g., air pollution or lack of green spaces, can also contribute to the development of chronic disease. Once vulnerable, some factors, e.g., high ambient temperatures, are thought of as being able to directly trigger an event as well (Chen et al., 2019). In light of these considerations, it becomes clear that predicting the exact timing of MI events is complex, as they are generally the result of developments that can span decades, after which they might be triggered randomly or as a result of otherwise benign circumstance, e.g., physical exercise, emotional stress or exposure to air pollution or heat stress during summer time.

However many of the risk factors that we cover in this study can still modify the individual likelihood of suffering from MI. Moreover, our research motivation is to eventually estimate the long-term tendencies in MI due to climate change. In light of this, we do not expect for the models to be able to accurately provide predictions on a daily basis. Instead, we decided to aggregate our model results on an annual basis. This should allow for some of the inherent randomness to average out and allow a more statistical view on the developments.

In the next Section 2, we present the methods used to develop the ML models. In Section 3, we describe the process to derive the input for our data-driven approach. In Section 4 the results of the simulations and their performance are given. In Section 5 we discuss the results and give an outlook for using the models to project future MI events, and finally in Section 6 we provide the conclusions.



## 2 Methods

In this section, we present the approach to modelling the occurrence of MI events from a large variety of data and discuss the ML methods that were applied. We also consider correlations among the features and describe how we selected suitable parameters for the ML algorithms.

### 2.1 A supervised learning problem for MI events

There are many ways to build ML models from data, e.g., classification or regression based algorithms. In this study, we focus solely on regression methods, as the registry data is case-only, i.e., by design each participant is bound to have an MI. It is therefore not possible to build individual risk profiles or estimate the likelihood of suffering MI, due to the lack of a healthy control group. The data, however, constitutes a very good basis for predicting the number of expected MI events with regression.

The target variable, i.e., the data one is interested in learning from or about in our case is the time series of daily events of MI observed in the study region. In addition, many of the co-occurring environmental variables that have a plausible causal relation to this target variable are collected and used as predictors in the training process.

We apply a range of different ML algorithms in order to test the performance of different learning approaches for predicting MI events based on the data. We use the scikit-learn package for performing the calculations (see Pedregosa et al., 2011; Pedregosa, F. et al.). The figures were produced using disability-friendly colormaps (Crameri et al., 2020). Before discussing the individual regression methods that are applied to the problem, we first describe how the supervised learning problem is posed based on the daily time series of MI events and the corresponding predictive variables. Here, we use the sliding window method.

For any given day $d$ let $y_d$ be the number of MI events and $x_{i,d}$ the value of the $i$-th predictive variable on that day (e.g., daily maximum temperature or daily mean $PM_{10}$). To work with standard regression algorithms a fixed number of features must be selected and together with the target value $y_d$ be provided as training input. The variables $x_{i,d}$ represent a time series and therefore only a subset of them should be selected as a feature of the regression problem, namely the conditions on the day of prediction. Past conditions, however, might also have an influence on current events, both long and short term. The sliding window method allows for this by selecting the features with a lag $n$, referred to as the window size. The merits of allowing for shorter or longer memories are difficult to estimate. For instance, the effects of extremely high temperatures on MI are generally expected to be short-term (Breitner et al., 2014), ranging from immediate effects to up to two weeks lag. The vector of features, i.e., the training (or test) instance on day $d$, is then given as:

$$x_d = (x_{1,d-n+1}, x_{1,d-n+2}, \dots, x_{1,d}, x_{2,d-n+1}, \dots, x_{2,d}, x_{m,d-n+1}, \dots, x_{m,d})$$

where $n$ is the windows size and $m$ the number of variables. Each predictive variable then yields $n$ features and the total number of features for this problem is $n \cdot m$. Accumulating the $x_d$ and $y_d$ for all days into a matrix $X$ and a vector $y$ yields input that can directly be used with the scikit-learn regression algorithms. We applied the ML methods and associated scikit-learn



classes listed in Table 1, together with their abbreviations as used in the remainder of this paper. Note that some features, such as the slowly changing demographic variables, were not subject to the sliding window and instead simply used the value on the day of prediction. For this study, after testing different lags between 1 and 14 days, we exclusively used a lag of $n = 3$ days as this resulted in the best overall scores.

Note that throughout this paper, we use the terms predictor and feature in an interchangeable manner, namely to refer to the features of the supervised learning problem derived above (the vector $X$ and its components). The term variable is used when referring to data that is not (directly) used as a feature of the learning problem, e.g., MI counts, or when addressing predictors in a more general manner. For instance, temperature is a variable used in this study, but it is to be differentiated from the mean temperature feature TMK. This differentiation is important as the features depend on the chosen window size, i.e.,

a window size of $n = 3$ results in three individual features, that represent the mean temperature on three different days during and preceding the MI event.

In addition to the environmental and socio-demographic predictors, we also added a random feature to be able to use its importance as a benchmark. Predictors less important than the random feature can be assumed to be irrelevant.

| Regression Method | Abbreviation | scikit-learn Class | Version |
|---|---|---|---|
| Decision Tree | DTR | sklearn.tree.DecisionTreeRegressor | 0.23.2 |
| Random Forest | RF | sklearn.ensemble.RandomForestRegressor | 0.23.2 |
| Gradient Boosting | GBR | sklearn.ensemble.GradientBoostingRegressor | 0.23.2 |
| Ridge Regression | RR | sklearn.linear_model.Ridge | 0.23.2 |
| Multi-layer Perceptron | MLP | sklearn.neural_network.MLPRegressor | 0.23.2 |

**Table 1.** Regression methods used and associated scikit-learn classes.

## 2.2   Scaling and random split

Different magnitudes of the features can have adverse effects as the results could be biased towards those variables given in nominally large units relative to others. To avoid this, we apply the sklearn.preprocessing.StandardScaler class to the input, resulting in features that are centered around 0 with unit variance. Second, we withhold parts of the data from the training to have independent data instances for validation. We apply sklearn.model_selection.train_test_split with shuffle, resulting in a random 75%/25% split of the data in training and test portions. The 25% of data not used for training the algorithms are used

for validation. Splitting the data randomly means that the underlying time series lose their natural temporal order. This has implications when visualising and interpreting model results that we will cover in a later section, but it reduces the likelihood of autocorrelations (e.g., seasonal signals) present in the time series to result in overoptimistic predictions. Note that in order to split the data randomly, the random number generator has to be initialised with a seed. We found that different random seeds can result in significantly different results. To avoid reporting results that are strongly dependent on the chosen seed, we

repeated all calculations with 100 randomly selected seeds. The result with the $R^2$-score closest to the average score of the





ensemble was then selected as a representative example of model capability. Moreover, as the dependency on the random seed is likely related to unbalanced splits, we employed a simple stratification strategy. The data is stratified along the number of MI occurrences observed, i.e., data points with the same number of MI are split among test and training in a representative way. This is especially important for rare events, such as 5 or more MI in one day. The dependency on the random seed was

substantially reduced in this way, but significant differences between different seeds could still be observed.

## 2.3 Feature reduction

A common concern in the application of ML algorithms is the influence of correlated features, where redundancy in the related variables can lead to an overemphasis of their influence on the target variable. This can be counteracted by choosing only one of the correlated features, usually the one that has the strongest correlation with the target variable. In our case,

we aimed to include as many variables as possible that could reasonably have an effect on MI. The downside is that some features, for instance maximum, minimum, and mean temperature, are highly correlated on a daily basis. A visualisation of the correlation between the predictors used in this study is shown in appendix Figure A8. To address this issue we tested the option of transforming the data to a smaller feature space using principal component analysis (PCA). The resulting principal components are uncorrelated to each other and the risk of introducing spurious or overly strong relationships into the training

data is reduced while retaining most of the original information. We used sklearn.decomposition.PCA and opted to retain at least 98% of the variance. Having the principal components as optional features allowed us to compare predictions with PCA to estimate the potential adverse effects of correlations present in our data. The results using the PCA data (not shown here) did not improve, suggesting that using the original set of features does not introduce spurious relations. Moreover, using PCA leads to a reduction of interpretability, as the principal components are linear combinations of the original features, without a

clear relation to the original variables.

## 2.4 Hyperparameter optimisation

The ability of the ML algorithms listed in Table 1 to produce accurate predictions is dependent on the selection of appropriate hyperparameters. These parameters generally control specific aspects of the underlying methods, such as the maximum depth of a decision tree, the number of neurons in a layer, or the strength of regularisation. With regularisation, a penalty is added as

model complexity increases, which helps to avoid overfitting. In this study, we used the sklearn.model_selection.GridSearchCV class to optimise hyperparameters over predefined parameter spaces with 5-fold cross validation. For this study we used the $R^2$ as the governing score to make decisions on optimal parameters. Repeating the procedure for all five possible splits and averaging the computed scores yields the final score for the given parameter values. Iterating over the hyperparameter space the parameter set with the best overall score is selected. Using cross-validation allows to produce more robust generalisation

error estimates without having to reserve a dedicated cross-validation set that would not be available for training. Moreover, by using folds based only on 75% of the training data, no information from the remaining 25% data is used for optimising the models and validation through parameter selection, i.e., leakage is prevented.

Ideally, a very comprehensive subset of the available hyperparameters would be optimised over a large and dense parameter





space. In practice, time and available computing resources limit the extent to which an exhaustive optimisation can be carried
out. Due to the curse of dimensionality, the number of possible combinations of parameters to test can easily become pro-
hibitively. This applies even when the individual training process is rather fast and makes the optimisation too time-consuming
to compute for practical purposes. We therefore performed a partial optimisation to circumvent this issue, i.e., we optimised
over rather sparse parameter spaces and a limited number of the available parameters.

Here, some of the hyperparameters are chosen to optimise the models based on their purpose, the importance of what they
control, and prevalence between the methods present in this paper (see Table 2). The listed parameters are used as model
optimisation variables simultaneously. As an example, in tree-based methods (DTR and RF) the selected parameters include
the maximum depth of the tree as well as splitting criteria for internal nodes.

On the other hand, the procedure limits the extent to which the exponential growth of the parameter space with increasing
numbers of parameters or increasing density results in prohibitive execution times. This makes the workload more manageable
and works as a trade-off between accuracy and feasibility. Table 2 shows a list of the selected hyperparameters for all the
methods used as well as their optimised values.

Fortunately, for most of these models there are parallel implementations of their algorithms to speed up the process. Here,
we used the Intel® extension package for scikit-learn, called scikit-learn-intelex. It has been designed to dynamically patch
scikit-learn estimators making it much faster to optimise the models used in this study with minimal changes to the code. It
is important to note, however, that the extension does not enhance all scikit-learn algorithms, such as the gradient boosting
regressor.

| Hyperparameters | random_state | max_depth | max_features | min_sample_leaf | min_sample_split | n_estimators | alpha | learning_rate | max_iter | activation | parallel |
|---|---|---|---|---|---|---|---|---|---|---|---|
| DTR | 429 | 10 | log2 | 64 | 2 | - | - | - | - | - | no |
| RF | 371 | 5 | log2 | 2 | 5 | 100 | - | - | - | - | yes |
| GBR | 357 | 3 | log2 | 64 | 2 | 100 | - | - | - | - | no |
| RR | 0 | - | - | - | - | - | 5000.0 | - | - | - | yes |
| MLP | 419 | - | - | - | - | - | 5.0 | constant | 2500 | logistic | yes |

**Table 2.** Optimised hyperparmeters marked with grey cells

## 3 Data

The dataset used in this study is highly heterogeneous along many dimensions, with differences ranging from file format,
metadata conventions, spatial coverage (e.g., regional, local) and resolution, to temporal frequency (e.g., daily, monthly, annual)
and representation (e.g., raster, polygon and point data). The capabilities of ML algorithms strongly depend on the quality of the
data used. In this section, we give an overview of the data used in this study and describe the workflow applied to homogenise
and prepare these. Table 3 lists all environmental and demographic predictive variables that were used for this study in addition
to the MI data and their characteristics, as well as the source datasets and associated references.





| Variable | Abbreviation | Unit | Time period | Resolution | Dataset | Reference |
|---|---|---|---|---|---|---|
| MI events (DONSET) | MI | # | 1985-2015 | District, daily | KORA | Helmholtz Zentrum München |
| Maximum temperature | TXK | °C | 1985-2015 | Station, daily | DWD | Deutscher Wetterdienst (DWD) |
| Mean temperature | TMK | °C | 1985-2015 | Station, daily | DWD | Deutscher Wetterdienst (DWD) |
| Minimum temperature | TNK | °C | 1985-2015 | Station, daily | DWD | Deutscher Wetterdienst (DWD) |
| Relative humidity | UPM | % | 1985-2015 | Station, daily | DWD | Deutscher Wetterdienst (DWD) |
| Apparent temperature | ATMK | °C | 1985-2015 | Grid, daily | This study | - |
| Maximum apparent temperature | ATXK | °C | 1985-2015 | Grid, daily | This study | - |
| Minimum apparent temperature | ATNK | °C | 1985-2015 | Grid, daily | This study | - |
| Vegetation index | NDVI | - | 1998-2015 | 1 km2, 10-daily | NDVI v2 | Copernicus Global Land Service (CGLS) |
| Nitrogen oxide | NO | ppm | 1993-2015 | Station, daily | LÜB | Bayerische Landesamt für Umwelt |
| Nitrogen dioxide | $NO_2$ | ppm | 1993-2015 | Station, daily | LÜB | Bayerische Landesamt für Umwelt |
| Sulfur dioxide | $SO_2$ | ppm | 1980-2015 | Station, daily | LÜB | Bayerische Landesamt für Umwelt |
| Ozone | $O_3$ | ppm | 1990-2015 | Station, daily | LÜB | Bayerische Landesamt für Umwelt |
| Particulate matter | $PM_{10}$ | ppm | 1980-2015 | Station, daily | LÜB | Bayerische Landesamt für Umwelt |
| Male population (total) | mtotal | # | 1985-2015 | District, annual | LfStat | Bayerisches Landesamt für Statistik |
| Female population (total) | ftotal | # | 1985-2015 | District, annual | LfStat | Bayerisches Landesamt für Statistik |
| Male/female population age under 1 | u1m, u1f | # | 1985-2015 | District, annual | LfStat | Bayerisches Landesamt für Statistik |
| Male/female population age 1 to 4 | 1t4m, 1t4f | # | 1985-2015 | District, annual | LfStat | Bayerisches Landesamt für Statistik |
| Male/female population age 5 to 9 | 5t9m, 5t9f | # | 1985-2015 | District, annual | LfStat | Bayerisches Landesamt für Statistik |
| Male/female population age 10 to 14 | 10t14m, 10t14f | # | 1985-2015 | District, annual | LfStat | Bayerisches Landesamt für Statistik |
| Male/female population age 15 to 19 | 15t19m, 15t19f | # | 1985-2015 | District, annual | LfStat | Bayerisches Landesamt für Statistik |
| Male/female population age 20 to 24 | 20t24m, 20t24f | # | 1985-2015 | District, annual | LfStat | Bayerisches Landesamt für Statistik |
| Male/female population age 25 to 29 | 25t29m, 25t29f | # | 1985-2015 | District, annual | LfStat | Bayerisches Landesamt für Statistik |
| Male/female population age 30 to 34 | 30t34m, 30t34f | # | 1985-2015 | District, annual | LfStat | Bayerisches Landesamt für Statistik |
| Male/female population age 35 to 39 | 35t39m, 35t39f | # | 1985-2015 | District, annual | LfStat | Bayerisches Landesamt für Statistik |
| Male/female population age 40 to 44 | 40t44m, 40t44f | # | 1985-2015 | District, annual | LfStat | Bayerisches Landesamt für Statistik |
| Male/female population age 45 to 49 | 45t49m, 45t49f | # | 1985-2015 | District, annual | LfStat | Bayerisches Landesamt für Statistik |
| Male/female population age 50 to 54 | 50t54m, 50t54f | # | 1985-2015 | District, annual | LfStat | Bayerisches Landesamt für Statistik |
| Male/female population age 55 to 59 | 55t59m, 55t59f | # | 1985-2015 | District, annual | LfStat | Bayerisches Landesamt für Statistik |
| Male/female population age 60 to 64 | 60t64m, 60t64f | # | 1985-2015 | District, annual | LfStat | Bayerisches Landesamt für Statistik |
| Male/female population age 65 to 74 | 65t74m, 65t74f | # | 1985-2015 | District, annual | LfStat | Bayerisches Landesamt für Statistik |
| Male/female population age over 75 | o75m, o75f | # | 1985-2015 | District, annual | LfStat | Bayerisches Landesamt für Statistik |
| Random variable | RND | fractional | 1985-2015 | Study area, daily | This study | - |

**Table 3.** Overview of predictive variables, source datasets and their origin.

## 3.1 KORA MI registry

The health dataset for our study is the KORA/MONICA MI Registry (see Tunstall-Pedoe et al., 1994; Holle et al., 2005), comprising records of MI events that occurred within the study region from 1985 to 2015. These data were collected at the hospitals in the Augsburg region. Each record contains the date of the MI occurrence, age and sex of the patient. Depending on availability complementary information is given, such as the patients' residential county (Landkreis), their body mass




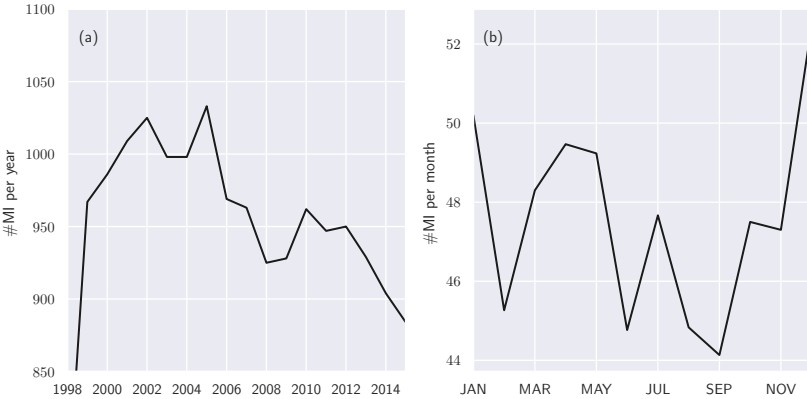

**Figure 1.** Number of annual MI (a) and mean annual cycle (b) in people aged under 75 from 1985 to 2015 for the study region (city of Augsburg and counties Aichach-Friedberg and Augsburg).

index (BMI), smoking status, and preexisting conditions such as diabetes. Although no detailed information is provided on

the location of an MI event, they can be assigned to either the urban (City of Augsburg) or one of the two rural counties (Landkreise) of the study region (Landkreis Augsburg and Aichach-Friedberg). As pointed out earlier, the individual, i.e., participant-specific information, could not be used as predictive data due to the nature of the regression approach, which aims to predict the gross number of MI in the population, rather than the outcomes of specific individuals. It is, however, possible to use this data to confine investigations to subgroups, e.g., to inhabitants of either urban or rural areas, and also to the elderly

or smokers, albeit at the cost of being limited to a smaller subset of the overall data. In total the number of recorded MI is $n = 34618$. Until 2008 the study was limited to participants of up to 74 years of age, with $n = 30081$ records total in that category. Figure 1 shows the aggregated number of MI per year and the mean annual cycle for the population aged under 75. The yearly maximum in MI is observed during the winter months, whereas the summer time shows the lowest occurrences. To generate the ground truth for our regression problem, we counted the total daily number of MI observed in the KORA study

and used the resultant time series as input for the ML algorithms.

## 3.2   Air temperature and humidity

Given the research goal of constructing relations between extreme temperature events and MI events, air temperature close to the ground is the most important factor to consider as the most direct measure of human exposure to heat and cold. The relatively small spatial scale of the study region (1998 km$^2$) puts high demand on the data in terms of resolution and accuracy.

At the same time, daily environmental data are required for our approach.

We opted to derive a 1x1 km grid for the study period between 1985 and 2015 from daily DWD station data in the vicinity of Augsburg and its neighboring districts. To this end, we applied universal Kriging with linear drift to the daily values at the temperature stations shown in Figure 2. The resulting gridded datasets (minimum, maximum and mean temperature) were





aggregated to the counties comprising the study region. This relatively simple approach proved to be accurate enough to obtain
realistic aggregated daily time series for the study region, as shown by the reasonable predictions in this paper.

We also include humidity related features in the models to gauge their relative importance in the models, despite the fact
that humidity is often considered less important. For this project only relative humidity from the DWD was available. We
applied the same Kriging procedure as used for temperature. To account for possible effects of perceived heat stress expressed
by simultaneous high humidity and high temperatures we included apparent temperature. Measures of apparent temperatures
relate a given temperature to the ambient humidity to account for the perceived temperature differences between dry and
humid conditions. Here, we applied a formula by Davis et al. (2016) to derive the apparent temperature based on temperature
and humidity data. The specifics of the computation can be found in the Appendix A1.

In a next step, the data was aggregated for the three different counties within the model region, the urban and the two rural
areas by computing weighted area means. The resulting daily time series can be readily used as input to the machine learning
models, as described in Section 2.

### 3.3 Air quality

Air pollution is complex and several particulate and gaseous pollutants should be considered in investigating its effects on MI
(e.g., Chen et al., 2018; Bourdrel et al., 2017). From the "Bavarian Air Hygiene State Monitoring System" (LÜB) database
(Bayerische Landesamt für Umwelt) we collected data on $PM_{10}$, NO, $NO_2$, Ozone ($O_3$), and $SO_2$ concentrations at multiple
stations across Bavaria in daily resolution.

Table A1 in the Appendix gives an overview of the selected measuring stations and their urban-rural categories, the corresponding pollutants data and their availability. Figure 2 gives an overview of the selected temperature and air quality measurement stations. We determined the aggregated daily means by calculating the mean values of the aforementioned stations,
taking into consideration their location proximity to the city centers, traffic-loaded inner-city streets, on industrial areas, on the
outskirts, or the large-scale background pollution. Such hot spots of pollution should have large effect on the calculated daily
mean.

The map indicates that there are only few air quality stations within the study region (five blue circles, and an archived station
with red border circle). Moreover, these stations have not all been always active during our study period. In this case we use
merely the active stations. However, if none of the regularly used station in the counties had recorded data on a given day, espe-
cially for the surrounding counties, alternative stations (light blue dots in Figure 2) from outside the study region were used as
replacements for the calculations. This has been achieved through an acceptable 10-15 percent error criterion for the monthly
value of alternative stations compared to the calculated monthly mean value of the county over a span of time provided by the
monitoring system. Obviously, the alternative stations should also have the same proximity setting. The calculated monthly
mean time series have been provided in the appendix figure A9.





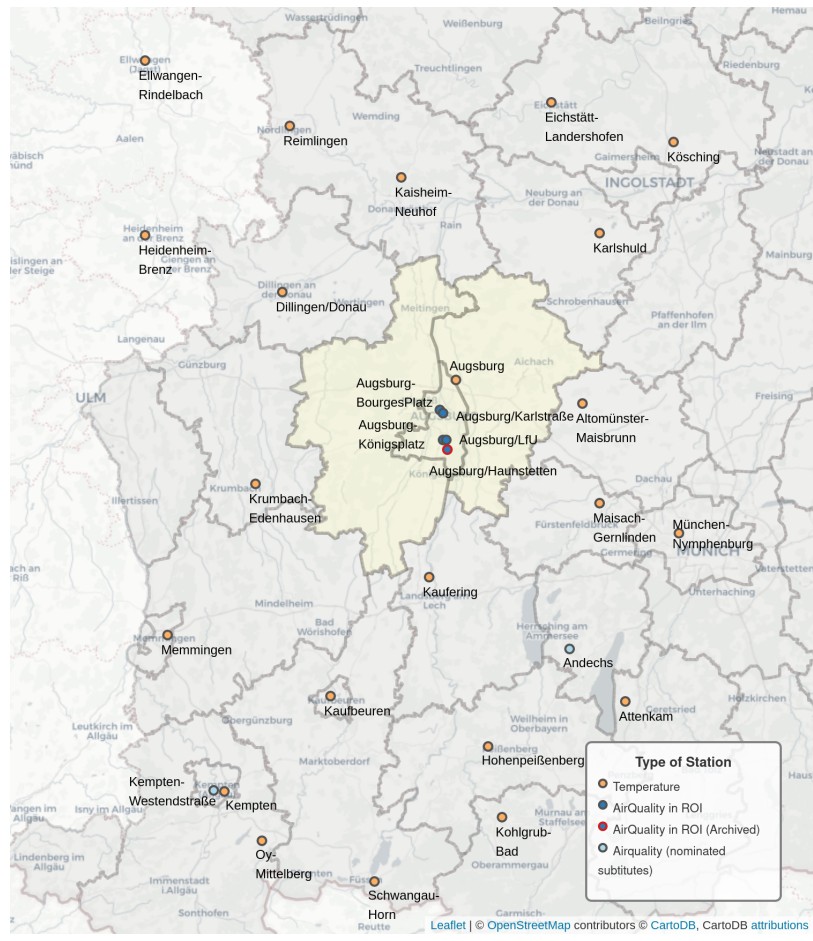

**Figure 2.** Air quality (blue) and temperature (orange) stations in the region of interest (ROI) around Augsburg.

## 3.4 Vegetation

The Normalised Difference Vegetation Index (NDVI) is an indicator of the greenness of the natural vegetation and other vegetation types such as agriculture and parks and gardens. It is widely used for ecosystems monitoring. In this study NDVI also is used as a proxy for shade as well as potential local cooling effect of vegetation by absorbing sunlight and through evapotranspiration. The NDVI_v2_1km database of the Copernicus Global Land Service (CGLS) vegetation products is freely available at a 1x1 km spatial resolution starting on April 1998 measured every ten days. We extracted the NDVI in our region of interest and a cubic spline interpolation has been used to upscale the temporal resolution from 10 days to daily values. Given the very gradual rate of change in vegetation cover and consequently the NDVI, we assume this interpolation does not produce large errors. Note that due to lack of availability of NDVI data before 1998, training and testing of the algorithms had to be confined to the time between April 1998 and December 2015.



## 3.5 Demographics

The absolute number of MI does not only depend on various environmental risk factors but also on the size and characteristics of the population. Disregarding other factors, any change in the absolute number of inhabitants would produce a similar change in the number of cases of MI as well. Moreover, both age and sex are strongly correlated with health outcomes in general, and specifically so for MI. Given trends of increasing urbanisation and rural depopulation as well as that of an ageing society, it is important to account for changes both in number of inhabitants and age stratification of the population in time. In addition, domestic migration reflected in relative changes between urban and rural parts, leading to differential changes in exposure to environmental hazards in the Augsburg region, can be important as well. We collected data from the Bavarion Office of Statistics. The data comprises annual values for the total number of inhabitants for each of the three counties, as well as the distribution of sex and age in the population from 1985 to 2015. Overall, 17 different age groups are accounted for as listed in Table 3. Since the algorithms require daily input values, a linear interpolation was applied to estimate the development within a given year.

## 4 Results

### 4.1 Weekly predictions of MI events

Our models produce daily predictions of MI events based on the environmental and demographic features within the given window size. We found that the models are not able to reproduce the daily variability of MI with sufficient accuracy. As an example we show the daily predictions aggregated to 7-day intervals to increase visibility in Figure 3. The resultant scores are given in Table 4 for both training and validation respectively.

| Method | DTR | | RF | | GBR | | RR | | MLP | |
|---|---|---|---|---|---|---|---|---|---|---|
| Score | Train | Test | Train | Test | Train | Test | Train | Test | Train | Test |
| Adj $R^2$ | 0.03 | -0.26 | 0.25 | -0.25 | 0.31 | -0.33 | -0.01 | -0.24 | 0.03 | -0.19 |
| $R^2$ | 0.1 | 0.03 | 0.31 | 0.04 | 0.36 | -0.03 | 0.06 | 0.04 | 0.1 | 0.09 |
| Max-Error | 13.66 | 14.31 | 13.5 | 14.73 | 12.77 | 14.8 | 14.84 | 14.87 | 14.97 | 13.95 |
| RMSE | 4.43 | 4.44 | 3.89 | 4.43 | 3.73 | 4.58 | 4.52 | 4.42 | 4.42 | 4.32 |
| BIC | 2416.08 | 986.01 | 2237.21 | 985.18 | 2176.49 | 999.99 | 2444.23 | 983.6 | 2414.36 | 972.77 |

**Table 4.** Training and test scores for 7-day aggregated daily predictions.

Although the seven-day predictions fit reasonably well for the training period, for the testing period the models do not predict 7-day variations (or day-to-day predictions) accurately enough. The predictions are too close to the mean and lack the variability displayed by the observations. This is likely related to randomness as well as risk factors that affect MI events that were not considered in the models. For instance, the temperature or air quality predictors may not sufficiently capture actual

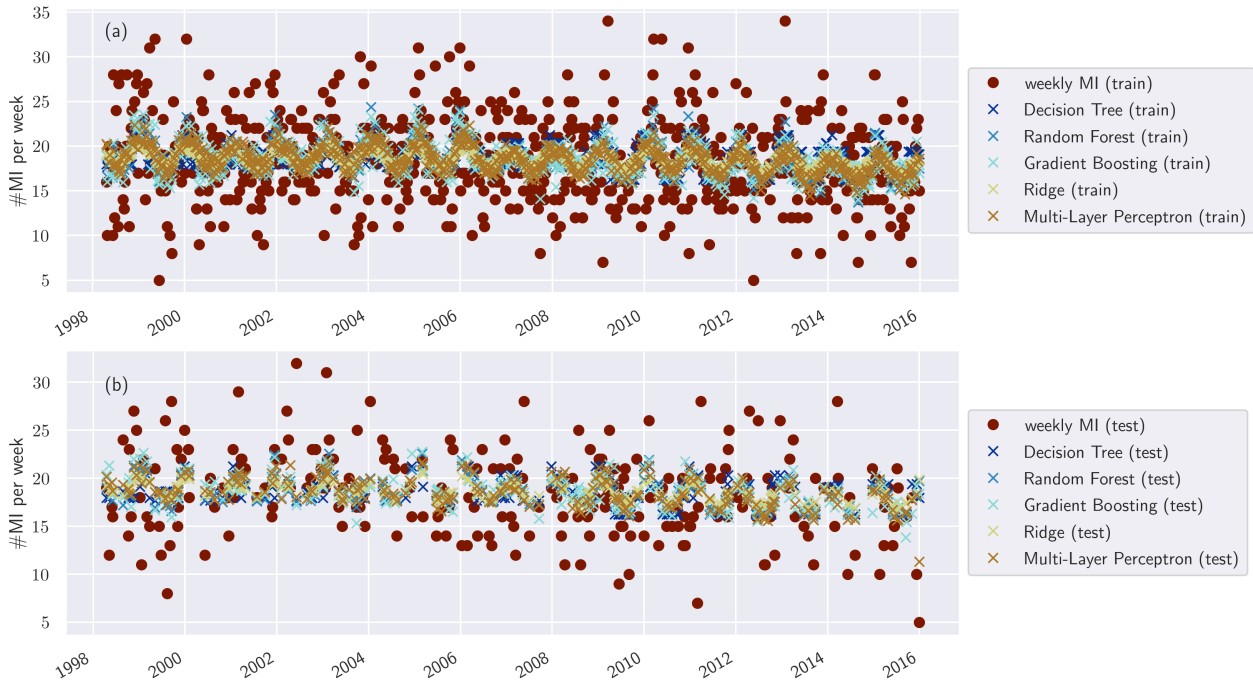

**Figure 3.** Daily predictions aggregated to 7-day intervals for all models. Shows predicted (crossed) and observed MI (dotted) for the training (top) and test (bottom) sets.

local circumstances, but also information about the built environment and other conditions that can not be easily accounted for is missing.

## 4.2 Annual predictions of MI events

Figure 4 shows the model performances on both the training and test sets as well as the actually observed MI as a reference for the five ML models given in Table 1. After training the models and performing the daily prediction on the test set, the results were aggregated to annual sums. By aggregating the model results to an annual basis, some of the inherent randomness is averaged out. Based on the annualised prediction results and time series of observed MI, the performance scores were derived (see Table 5). The training scores demonstrate that the ML models are able to predict the year-to-year variations quite well, with
adjusted $R^2$ scores between 0.86 and 0.93. The performance on the test dataset is relevant for assessing the generalisation error for previously unseen data. In contrast to the training data, the results on the test set are less but still reasonably accurate, with adjusted $R^2$ scores between 0.59 and 0.71, showing that inter-annual variations and long-term trends are largely captured. The RR and MLP models exhibit the best performance, showing that both well-tuned linear models as well as neural networks are able to simulate the relations between environmental conditions and MI events. The DTR shows the lowest overall performance
by comparison.



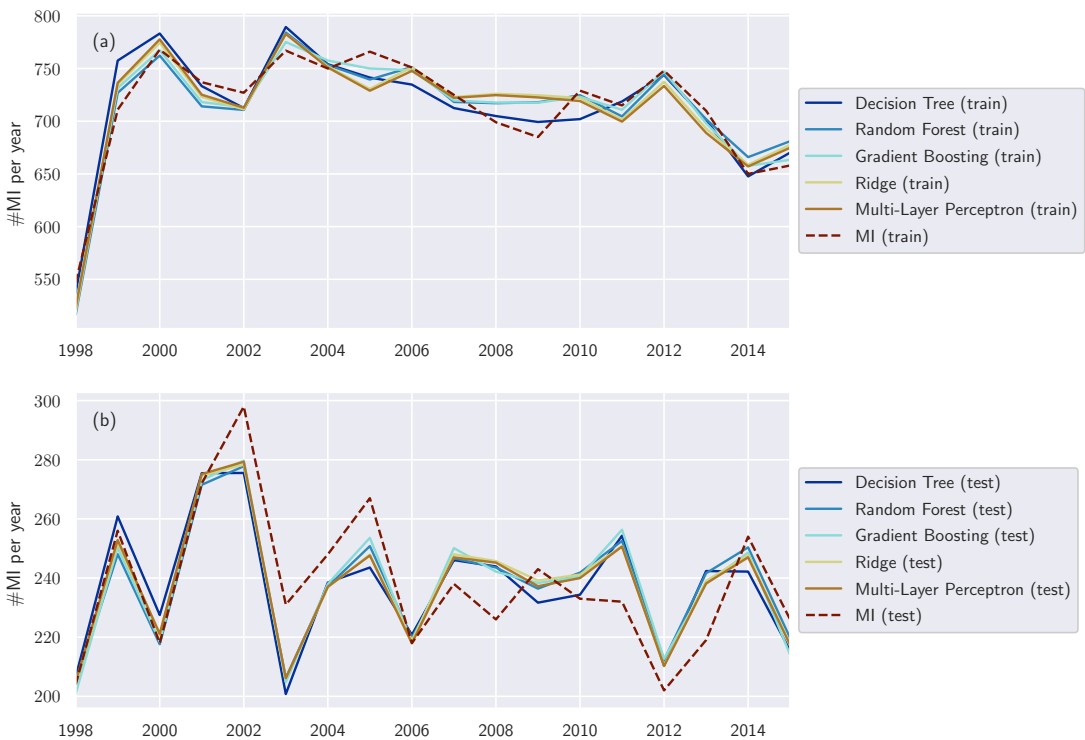

**Figure 4.** Annually aggregated predictions of MI in the general population for all models. Predicted (solid) and observed MI (dashed) for training (a) and test (b) sets.

| Method | DTR | | RF | | GBR | | RR | | MLP | |
|--------|-----|-----|-----|-----|-----|-----|-----|-----|-----|-----|
| Score | Train | Test | Train | Test | Train | Test | Train | Test | Train | Test |
| Adj $R^2$ | 0.89 | 0.59 | 0.89 | 0.69 | 0.93 | 0.69 | 0.86 | 0.7 | 0.86 | 0.71 |
| $R^2$ | 0.89 | 0.6 | 0.89 | 0.7 | 0.93 | 0.7 | 0.86 | 0.71 | 0.86 | 0.72 |
| Max-Error | 46.54 | 30.23 | 32.94 | 24.76 | 32.57 | 26.25 | 39.64 | 25.01 | 37.61 | 24.9 |
| RMSE | 17.49 | 15.06 | 17.41 | 13.05 | 13.98 | 13.07 | 19.34 | 12.98 | 19.3 | 12.76 |
| BIC | 259.1 | 253.71 | 258.93 | 248.56 | 251.04 | 248.6 | 262.72 | 248.37 | 262.64 | 247.76 |

**Table 5.** Training and test scores on annual basis for the general population.





### 4.3 Feature importance

In Figure 5 we show a condensed rendition of the feature importance where related variables have been grouped together of each model; except for the MLP which does not support feature importance within the scikit-learn framework. Note that variables subject to the sliding window were aggregated over the window length of three days to improve readability. Moreover,

features related to time such as the current year or the day of the week were also aggregated to a single group. More detailed plots retaining the differentiation of all features and window days can be found in the Appendix (see Figures A6 and A7). The latter Figure also shows that many of the original demographic features carry little to no weight. We therefore reduced the granularity of the demographic data to the age groups $0 - 29$, $30 - 49$, $50 - 74$ and $> 75$, generally yielding improved results.

While the performance of the models differs, some trends can be observed. Overall, the single most important group is air

quality, followed by temperature and demographic predictors. In contrast to apparent temperature humidity only exhibits low importance across all models. Time related features as well as NDVI exhibit the lowest explanatory power. NDVI is ranked very closely to the random feature by all models, and in some cases even slightly lower. All other features are consistently ranked above the random feature, indicating that they have relevance in predicting MI occurrence.

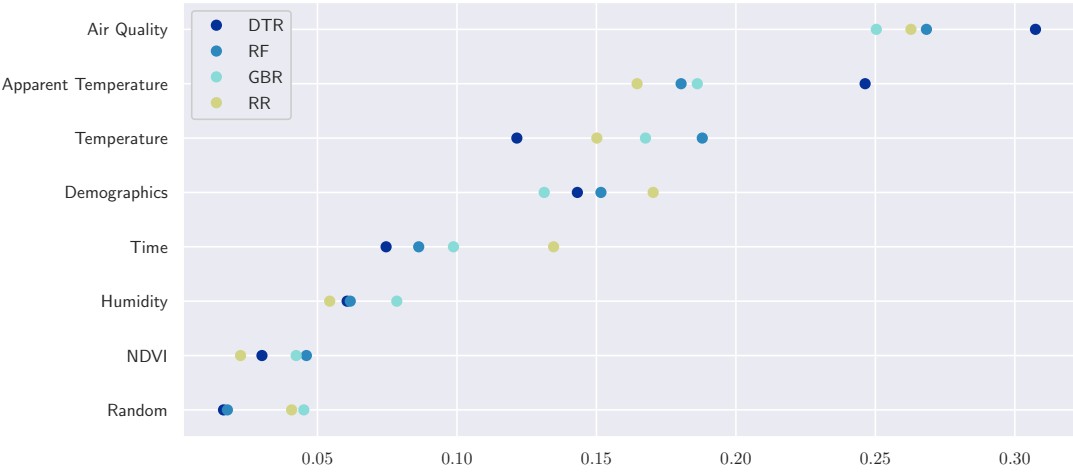

**Figure 5.** Aggregated feature importance for predicting MI for the general population. Related features have been grouped thematically. Larger values indicate higher importance and per model the sum over all features equals one.

### 4.4 Subgroup analysis

The models were also applied to subgroups of the population, albeit at the expense of a reduced amount of available training data (see Table 6) for an overview. For this analysis we selected a total of 5 subgroups: the urban (Augsburg city) and rural population (two adjacent counties) respectively, the elderly (people aged between 60 and 74), diabetics, and active smokers. The data was reduced to include only participants with the associated attribute. The training procedure was then repeated as




detailed for the general case on the resulting subsets. As expected, the validation scores dropped considerably for all subgroups,
likely a consequence of reduced amounts of training data. We refer to the Appendix for detailed results, but for the urban and
rural subgroups adjusted $R^2$ scores between $0.6$ and $0.37$ were observed in validation (see Tables A2 and A3). Both subgroups,
being of almost equal size, performed comparably well, with the urban population exhibiting slightly lower scores however.

| Population | n | | | Age | | | |
|---|---|---|---|---|---|---|---|
| | all | female | male | min | max | mean | stdev |
| General | 17134 | 4330 | 12804 | 25.0 | 74.0 | 62.34 | 9.42 |
| Urban | 8424 | 2248 | 6176 | 25.0 | 74.0 | 62.46 | 9.46 |
| Rural | 8710 | 2082 | 6628 | 25.0 | 74.0 | 62.22 | 9.39 |
| Elderly | 11470 | 3314 | 8156 | 60.0 | 74.0 | 67.94 | 4.26 |
| Diabetic | 5451 | 1521 | 3930 | 26.0 | 74.0 | 64.44 | 8.07 |
| Smoker | 3800 | 793 | 3007 | 26.0 | 74.0 | 55.72 | 9.76 |

**Table 6.** Overview of the number of cases as well as age and sex distribution for the different study populations considered.

The validation results for the elderly population (see Figure A3 and Table A4) are more accurate (adjusted $R^2$ between 0.5
and 0.68) than for the urban and rural populations, although the number of training samples is much higher in both of those
cases.

The results for the diabetic population are shown in Figure A4 and Table A5. As observed with the elderly, the scores for
the diabetic part of the population (adjusted $R^2$ between 0.44 and 0.55) are comparable to those of the (much bigger) rural and
urban subgroups, albeit slightly lower.

The results for the smoking population are shown in Figure A5 and the scores are given in Table A6. For this group adjusted
$R^2$ validation scores drop to around $0.42$ on average, indicating a less accurate fit than for all the other subgroups. This is
consistent with the smoker group being the smallest of the explored subgroups resulting in the lowest amount of training data
as well.

Overall, the skill of the models is clearly reduced when limited to subsets of the overall data. The decrease in performance,
however, is quite different between subgroups, especially when taking into account their relative sizes. A particularly inter-
esting question is whether the variable importance for any one subgroup changes substantially in comparison to the general
population. Figure 6 shows the difference in variable importance for each of the subgroups in relation to that of the general
population. To aid readability related features have been grouped again. Considerable differences between subgroups, models
and feature groups can be observed. For instance, three out of four models agree that air quality, humidity and demographics are
particularly important for predicting urban MI, while giving less weight to the temperature related features. The importance of
time related indicators reduced consistently over the general population. In most cases the importance of the random feature is
also reduced, indicating increased robustness of the results. NDVI exhibits mostly increased importance, but with considerable
inter-model spread.


For the rural population the results suggest slightly increased relevance of air quality, and a bit more pronounced increases in humidity and NDVI. Temperature and apparent temperature mostly align with the results for the general population. The demographic indicators are less relevant when compared to the general case, as are the time related features. The random feature is mostly ranked as less important.

For the elderly, the models are undecided on air quality, with a slight tendency towards reduced importance. On the other hand, the weight of the demographic features is emphasized in comparison to the general case. Less importance is also attributed to relative humidity and apparent temperature. On both temperature and time related features, as well as the random feature the models are undecided, whereas NDVI is mostly found to be more important.

For the diabetic population, the models mostly agree that demographic features and temperature are more important in predicting MI for this group in comparison to the general population. On the other hand, relative humidity, time related features and the random feature are ranked lower. NDVI is mostly considered to be more important overall. Only for the air quality group and the apparent temperature are the models undecided, with a tendency towards decreased importance in both cases.

Lastly, for the group of active smokers the models mostly suggest an increased importance of air quality as well as demographic features and NDVI for the prediction of MI. Relative humidity as well as apparent temperature and time-related features are overall considered less important. The models are undecided on temperature but with a tendency of increased relevance. The same holds true for the random feature, albeit with the opposite tendency.

## 5 Discussion

To our knowledge, this is the first study building and testing machine learning models that include more than only weather variables (such as Zhang2009 for heat mortality) for predicting MI incidence. The developed ML models have varying skill in predicting MI. At the daily to 7-day time scales, randomness seems too large to produce meaningful predictions. However, when the prediction are aggregated to annual sums, the models are very well capable of reproducing the inter-annual variability of observed MI as well as the long-term trends, also for the validation datasets. This is comparable to the performance of methods used for predicting malaria incidence (e.g., Sewe et al., 2017). In terms of performance scores the models achieve very similar outcomes both in training and validation (see Table 5), indicating some robustness of the predictions. More qualitative differences emerge, however, when investigating feature importance. There are substantial differences between the ML models in terms of some features (Figure 5). Most models rank air quality variations and temperatures among the most important features, but a large spread can be observed for humidity (UPM) and the time-related features. This indicates at least some inherent uncertainty. The feature importance for the MLP model, which had the highest performance, was not assessed in this study, due to limitations within the scikit-learn framework.

Classical epidemiological approaches like general linear or additive models are mostly interested in explaining the direction and corresponding uncertainty of an association between one or several (environmental) risk factors and health thereby adjusting for potential confounding factors. In case of potential non-linearities, the shape of the exposure-response curve is usually modeled as a smooth function. However, the models are limited in case of high correlation and/or high-dimensional

**Figure 6.** Change of feature importance for predicting MI in percent relative to the general population for every subgroup considered. Related features have been grouped thematically.

interactions between the covariates. The suggested ML approaches can (partly) handle these issues and offer the possibility to compare the importance and predictive performance of a multitude of environmental predictors.

The training scores in many cases are close to the maximum, with adjusted $R^2$ values greater than 0.85. This may be indicative of overfitting, possibly opening room for improving further on the generalisation by applying stronger regularisation. While the models were adjusted by optimising the hyperparameters, not all possible parameter values have been explored. For instance, in the case of the tree-based models pruning is an effective way to reduce overfitting. We did not apply pruning,





because this is a very time-consuming approach and we had to limit the extent to which we could perform an exhaustive optimisation. For the MLP and RR models regularising parameters were explicitly included in the optimisation, but possibly the ranges were not wide enough to achieve the best trade-off between training and validation.

The model results are sensitive to the selection of the random seed that is used in making the initial train-test split. We found that changes in the random seed routinely had greater impact than the choice of hyperparameters. One way of dealing with this would be to also include this random seed in the optimisation process. Currently, only the random seeds used for randomly selecting the folds in cross-validation and in initialising the regressors are optimised. In light of the strong influence of the initial split, however, we opted to instead test over a range of possible seeds and select the results closest to the average

performance of the models, not to overstate our results. The sensitivity to the initial split may indicate a lack of data, but is likely mostly due to unbalanced splits. We reduced this sensitivity by employing a simple but effective stratification strategy. This reduced the variation across seeds, but does not entirely resolve the issue. Possibly, more intricate stratification approaches may reduce the dependency even further.

    We were already able to indicate differences between different geographical regions, e.g. urban and rural populations. For

instance, humidity and air quality are more important predictors for the urban population, compared to the overall population. The models could be further improved by increasing the spatial representation, as the environmental predictors also would support this. Increased spatial representation would also allow for additional exposure metrics to be established and more predictors, such as those related to building structures, their insulation and energy efficiency.

    We used a sliding window of three days to allow for lagged effects. Depending on the variable, however, longer windows

might be more appropriate such as for cold exposure in winter. Ideally every predictor would have its own individual window length, derived as part of the optimisation process.

    An additional area of possible improvement is the environmental data. Some variables such as the NDVI and the air quality indicators were not fully available for the period between 1985 and 2015, effectively limiting the data available to the period from 1998 to 2015. Many variables were also derived from station networks after cleaning and applying a simple Kriging

method. It is possible that reducing bias in these data, for instance by using more sophisticated interpolation methods or additional data sources such as from remote sensing, could further improve the results.

    Finally, it could also be worthwhile to study the direction of the change related to a given feature as to get a better grasp on the qualitative relationships between predictors and MI events in the models. It should be noted, however, that other than RR all methods used are nonlinear and may therefore not exhibit simple directional effects.

All models consistently stressed the importance of air quality variables. Climate impact studies, especially related to MI, might therefore benefit from carefully analysing possible future developments of these variables. Electrification of traffic, reduction of fossil fuel and related changes might yield substantial improvements in air quality in the future. Instead of just focusing on projected changes in, e.g., temperature and humidity, scenarios for air quality need to be considered as well, to gain more precise insight into possible impacts of climate change on health and possible strategies to reduce these.

Current research and data availability in the fields of climate modelling, and demographic and environmental scenario development provide many opportunities to use the developed ML models from our research for projecting future health





risks. Ensembles of regional climate models provide climate projections with the highest spatial resolution. For the study region, EURO-CORDEX simulations (Jacob et al., 2014, 2020) can be considered as they provide the largest ensemble at a high spatial (0.11°, i.e., 12 km) and temporal (daily) resolution climate simulations that are available today. Several of the predictors used in this study could be derived from the EURO-CORDEX ensemble, namely temperatures and in many cases relative humidity, dew-point temperatures and therefore also apparent temperatures. Alternatively, an ensemble of convection permitting decadal regional climate simulations at roughly 3km, both for historic and future conditions, has been created within CORDEX FPS (e.g., Ban et al., 2021). Using an ensemble of near-future (2035-2065) climate model simulations allow for scenario uncertainty, internal climate variability, and climate model uncertainty to be assessed (Hawkins and Sutton, 2011) when comparing the changes in MI to the reference historical simulations.

Demographic predictions until 2039 for the study region at county level can be obtained from the Bayerisches Landesamt für Statistik. Longer-term projections up until the year 2060, albeit contingent on different socio-economic scenarios and at the level of the federal state of Bavaria, could be obtained from the Statistisches Bundesamt and be used to estimate the local projected demographics in the study area. These projections would provide a robust basis to estimate potential developments of the local population in the near-future.

Projections of vegetation changes, as represented by NDVI, at the required spatial scale are not readily available. On the other hand, it can be reasonably assumed that the potential for increased greenness in the inner city is limited. Likewise, the potential for substantial effects from added green in the rural surroundings of Augsburg is low, as it is already ubiquitous there. We therefore believe that moderate up- or downscaling of NDVI patterns observed in the past and present may suffice to yield suitable estimates of possible future developments, such as adaptation measures of increasing vegetation to reduce the urban heat island effect or offset impacts from overall warming due to anthropogenic climate change.

The air quality projections are related to the emission scenarios used by global climate models. For the CMIP6 climate models, estimates of regional surface air quality are available at the global model scale Turnock et al. (2020). These projections could be used to scale the observed daily air quality observations, but more exhaustive and local projection data would be preferred. To date, however, regional climate models do not feature the necessary complex chemical models to accurately model the transport, dispersion and diffusion of pollutants. Here, we also excite a pragmatic up- or downscaling (depending on related socioeconomic scenarios, e.g., fossil fuels vs. electrification of traffic) of the observed patterns to bridge the gap between the projections of future air pollution at the global level (from the global climate models) to the local level.

## 6 Conclusions

We have developed an approach for predicting MI events using multi-variable Machine Learning methods, based on environmental and demographic data for a case in Augsburg, Germany. Given that health outcomes depend on a multitude of factors, we applied this data-driven approach to establish relevant relationships, and collected several different variable datasets. We acquired heterogeneous data on MI events from the KORA MI registry, as well as weather, environmental and demographic data from various sources to create a meaningful and consistent daily time series of the predictive features and the target variable.





Starting from these time series, a supervised learning problem for MI was formulated, accounting for lagged effects of up to three days. Five different regression algorithms were trained on this data, based on random 75/25 train-test splits for the period between April 1998 and December 2015. An extensive effort has been carried out to optimise a selection of the various hyperparameters modifying the behaviour of the regressors, based on 5-fold cross validation with respect to the $R^2$ scores.

Applying the trained models on the unseen test data allowed an estimation of the generalisation error of the models. We found

that the daily predictions do not show meaningful predictions of MI events. We found that the annually aggregated predictions agree moderately well with the observed MI events, accurately reflecting observed trends and inter-annual variability of MI. The match between observations and the model predictions is supported by the observed validation scores, with adjusted $R^2$ scores ranging between 0.59 and 0.71. Overall, the models displayed comparable skill, but the RR and MLP models slightly outperformed the tree-based methods. The least accurate results were produced by the DTR model. Moreover, analysing the

feature importance where possible, we found that despite similar overall scores, the relative weight put into different features can vary substantially between the models. This emphasised the necessity to consider ensembles of models, as it allows to gauge the model spread and estimate inherent uncertainty. In this study, all models explored show that air quality is the most important feature to predict MI, followed by absolute and apparent temperatures, the latter including humidity. We also applied the models to various vulnerable subgroups, such as the elderly or diabetic patients, resulting in only slightly reduced skill

scores due to the reduced amounts of training data.

Possibilities to improve the current approach are manifold. One aspect is to improve on the quality of the predictive data, e.g., by using more advanced point to space methods rather than ordinary Kriging as was used in this study. Given the relatively good performance of MLP in this study more complex and potent neural networks could be explored to possibly improve on this further. Moreover, different ML approaches could be explored, such as density estimation and Bayesian methods, yielding

estimates of relative risk of different groups to suffer MI. Such estimates could be more readily compared with commonly used epidemiological models than the regression models presented here. Overall, the models' capacity to give reasonable estimates of possible future developments of MI based on the predictive features appears robust. In a next step we aim to apply the trained models to scenarios of future climatic, environmental and demographic conditions. This will allow estimating future changes in MI taking into account climatic, as well as other environmental and demographic changes that has most often not been

done. These changes could also include further improvements in air quality, or increased 'greening' of urban environments with vegetation. New estimates of changes in MI can be based on these ML methods, using ensembles of projected climate change, demographic scenarios including ageing population, and environmental scenarios contingent on societal transformation (e.g., electrification of traffic, greening of cities). Such estimates will enable to gauge the sensitivity of the complex health-environment interactions, and benefits of proposed environmental and health interventions in urban areas.





## Appendix A


### A1   Derivation of apparent temperature

We have computed the apparent temperature according to:

$$T_a = -2.653 + 0.994T + 0.01537T_d^2$$

where $T_a$ is the apparent temperature, $T$ the near-surface mean temperature and $T_d$ the near surface dewpoint temperature. Dewpoint temperature, however, was not available for this study. To facilitate estimating the apparent temperature we therefore first derived another humidity related quantity: vapour pressure. Applying again universal Kriging with linear drift, we arrived at 1x1 km gridded data for vapour pressure, applying the Magnus formula to estimate the dewpoint temperature:

$$T_d = \frac{b \cdot v}{a - v}$$

where $a = 7.5$, $b = 237.3$, $v = \log_{10}\left(\frac{p_v}{6.1078}\right)$ with $p_v$ the vapour pressure.

Applying these formulas to the gridded temperature and humidity data derived before yields a 1x1 km grid for apparent temperature. Note that the formulas were independently applied to mean, maximum and minimum temperature. Subsequent
aggregation over the model region then completed the preparation of apparent temperature as input feature.

### A2   Detailed subgroup and feature importance results

| Stations: | Augsburg/Königsplatz Time period(resolution) | Augsburg/BourgesPlatz Time period(resolution) | Augsburg/LfU Time period(resolution) | Augsburg/Karlstraße Time period(resolution) | Augsburg/Haunstetten Time period(resolution) | Andechs/Rothenfeld Time period(resolution) | Kempten/Westendstraße Time period(resolution) |
|---|---|---|---|---|---|---|---|
| Urban-Rural categories: | | | Urban | | | | Rural |
| $PM_{10}$ | 1980-1985 (daily) 1986-2004(3hr) 2005-2018(1hr) | 1986-2004(3hr) 2005-2018(1hr) 2013-2014 missing | 2000-2004(3hr) 2005-2018(1hr) 2010-2016 missing | 2003-2004(3hr) 2005-2018(1hr) | - | 2003-2004(3hr) 2005-2018(1hr) | 1980-1985 (daily) 1986-2004(3hr) 2005-2015(1hr) 2015-2018 missing |
| $PM_{2.5}$ | - | 2008-2018(1hr) | 2008-2018(1hr) 2010-2016 missing | - | - | 2012-2018(1hr) | 2014-2018(1hr) |
| $NO$ | 1980-2018(1hr) | 1986-2018(1hr) | 2000-2018(1hr) 2012-2013 missing | 2003-2018(1hr) | - | 2003-2018(1hr) | 1993-2018(1hr) |
| $NO_2$ | 1980-2018(1hr) | 1986-2018(1hr) | 2000-2018(1hr) 2012-2013 missing | 2003-2018(1hr) | - | 2003-2018(1hr) | 1993-2018(1hr) |
| Ozone $(O_3)$ | 1980-1985(1hr) | 2012-2018(1hr) | 2000-2018(1hr) | - | 1985-1999(1hr) | 2003-2018(1hr) | 1990-2018(1hr) |
| $CO$ | 1980-1999(1hr) only 2018(1hr) | 1987-1999(1hr) | only 2018(1hr) | only 2018(1hr) | 1980-1999(1hr) | - | 1980-1999(1hr) |
| $SO_2$ | 1980-2018(1hr) | 1986-2018(1hr) | 2000-2018(1hr) 2012-2013 missing | 2003-2018(1hr) | - | 2003-2018(1hr) | 1993-2018(1hr) |
| $BTX$ | 1980-2018(1hr) | 1986-2018(1hr) | 2000-2018(1hr) 2012-2013 missing | 2003-2018(1hr) | - | 2003-2018(1hr) | 1993-2018(1hr) |

**Table A1.** Air quality stations data availability and categories


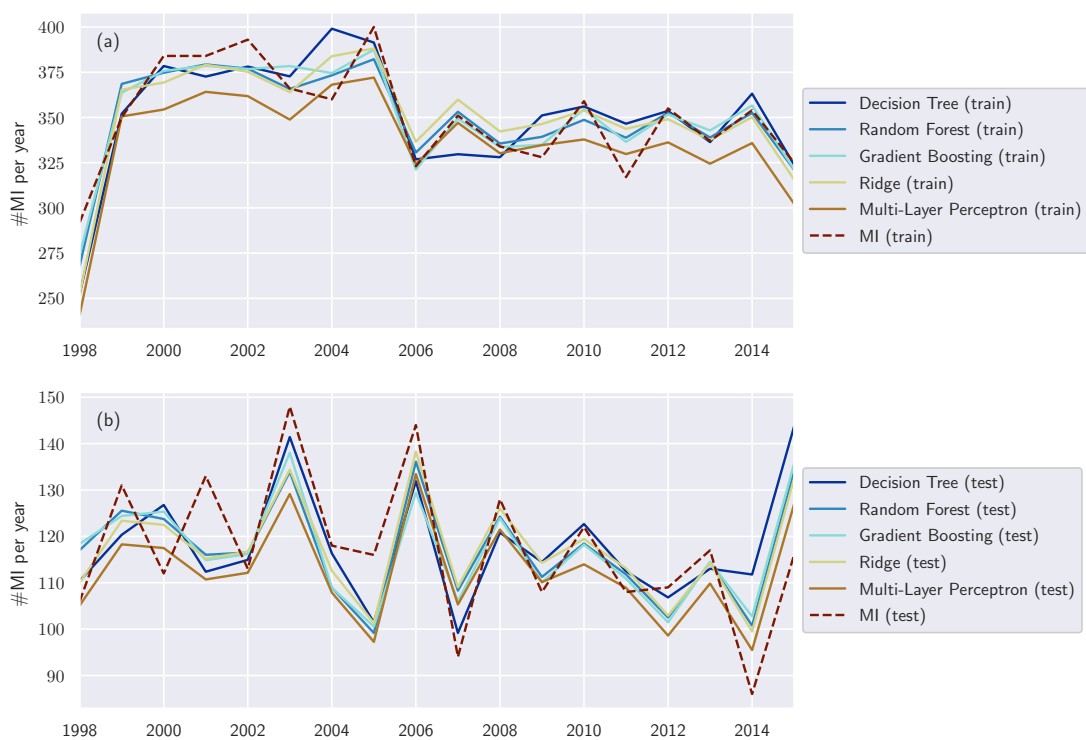

**Figure A1.** Annually aggregated predictions of MI in the urban population for all models. Predicted (solid) and observed MI (dashed) for training (a) and test (b) sets.

| Method | DTR | | RF | | GBR | | RR | | MLP | |
|---|---|---|---|---|---|---|---|---|---|---|
| Score | Train | Test | Train | Test | Train | Test | Train | Test | Train | Test |
| Adj $R^2$ | 0.59 | 0.31 | 0.81 | 0.49 | 0.86 | 0.44 | 0.66 | 0.56 | 0.41 | 0.45 |
| $R^2$ | 0.6 | 0.33 | 0.81 | 0.51 | 0.86 | 0.46 | 0.66 | 0.57 | 0.42 | 0.47 |
| Max-Error | 39.5 | 27.71 | 23.96 | 18.02 | 19.47 | 19.72 | 39.93 | 17.79 | 50.94 | 22.29 |
| RMSE | 17.64 | 12.42 | 11.97 | 10.71 | 10.28 | 11.22 | 16.07 | 9.94 | 21.12 | 11.11 |
| BIC | 259.4 | 246.79 | 245.44 | 241.43 | 239.98 | 243.12 | 256.04 | 238.77 | 265.89 | 242.76 |

**Table A2.** Training and test scores on annual basis for the urban population.



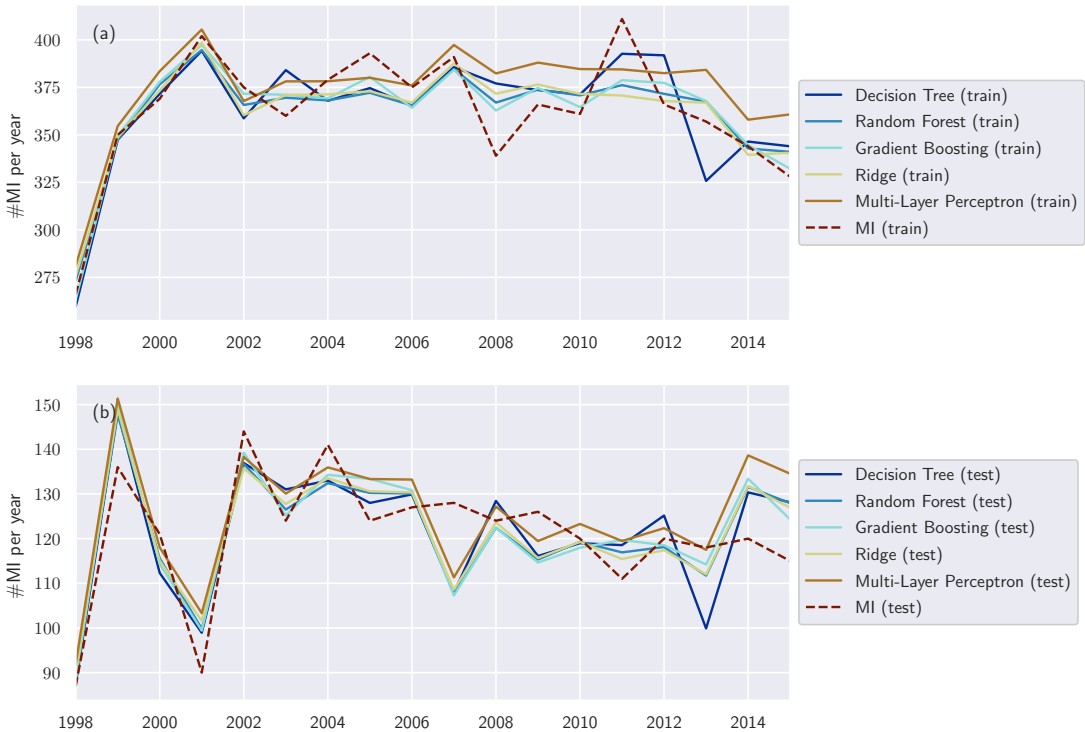

**Figure A2.** Annually aggregated predictions of MI in the rural population for all models. Predicted (solid) and observed MI (dashed) for training (a) and test (b) sets.

| Method | DTR | | RF | | GBR | | RR | | MLP | |
|--------|-----|-----|-----|-----|-----|-----|-----|-----|-----|-----|
| Score | Train | Test | Train | Test | Train | Test | Train | Test | Train | Test |
| Adj $R^2$ | 0.7 | 0.5 | 0.8 | 0.6 | 0.85 | 0.59 | 0.76 | 0.6 | 0.61 | 0.47 |
| $R^2$ | 0.7 | 0.52 | 0.8 | 0.62 | 0.85 | 0.6 | 0.77 | 0.61 | 0.61 | 0.49 |
| Max-Error | 38.07 | 20.12 | 34.81 | 20.36 | 32.2 | 20.72 | 40.29 | 19.45 | 43.32 | 19.53 |
| RMSE | 17.32 | 9.75 | 13.94 | 8.7 | 12.05 | 8.89 | 15.26 | 8.73 | 19.68 | 10.04 |
| BIC | 258.74 | 238.06 | 250.94 | 233.97 | 245.68 | 234.75 | 254.19 | 234.08 | 263.34 | 239.11 |

**Table A3.** Training and test scores on annual basis for the rural population.


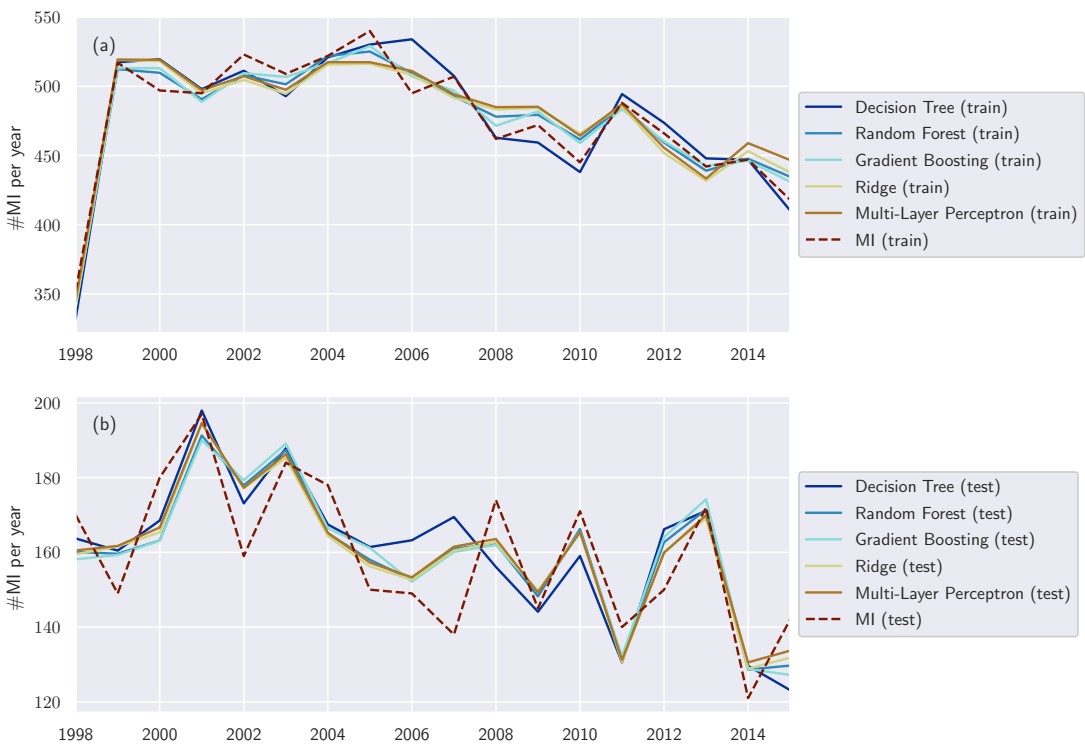

**Figure A3.** Annually aggregated predictions of MI in the elderly population for all models. Predicted (solid) and observed MI (dashed) for training (a) and test (b) sets.

| Method | DTR | | RF | | GBR | | RR | | MLP | |
|---|---|---|---|---|---|---|---|---|---|---|
| Score | Train | Test | Train | Test | Train | Test | Train | Test | Train | Test |
| Adj $R^2$ | 0.9 | 0.5 | 0.94 | 0.64 | 0.95 | 0.61 | 0.89 | 0.67 | 0.88 | 0.68 |
| $R^2$ | 0.91 | 0.52 | 0.94 | 0.65 | 0.95 | 0.62 | 0.89 | 0.68 | 0.88 | 0.69 |
| Max-Error | 39.0 | 31.46 | 16.6 | 22.98 | 16.12 | 22.1 | 23.71 | 22.48 | 28.67 | 23.51 |
| RMSE | 13.61 | 13.34 | 10.95 | 11.27 | 9.54 | 11.83 | 14.77 | 10.82 | 15.1 | 10.7 |
| BIC | 250.06 | 249.33 | 242.25 | 243.26 | 237.29 | 245.02 | 253.02 | 241.81 | 253.81 | 241.4 |

**Table A4.** Training and test scores on annual basis for the elderly population.




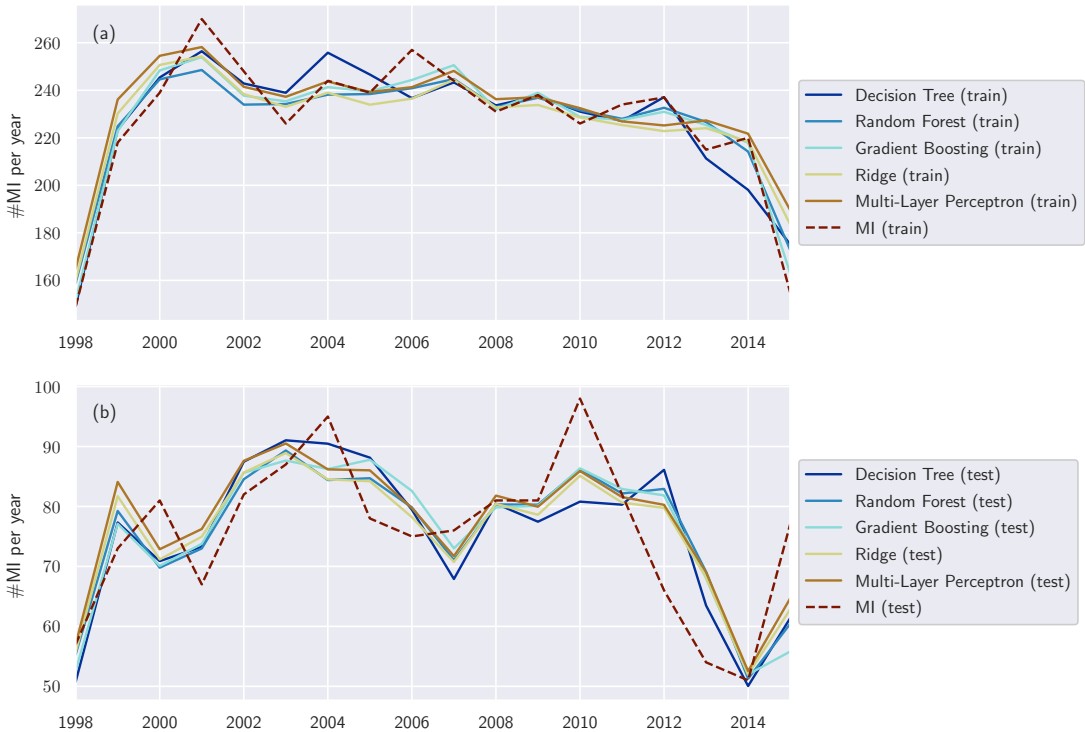

**Figure A4.** Annually aggregated predictions of MI in the diabetic population for all models. Predicted (solid) and observed MI (dashed) for training (a) and test (b) sets.

| Method | DTR | | RF | | GBR | | RR | | MLP | |
|--------|------|------|------|------|------|------|------|------|------|------|
| Score | Train | Test | Train | Test | Train | Test | Train | Test | Train | Test |
| Adj $R^2$ | 0.87 | 0.44 | 0.89 | 0.5 | 0.93 | 0.44 | 0.84 | 0.55 | 0.81 | 0.54 |
| $R^2$ | 0.87 | 0.46 | 0.89 | 0.51 | 0.93 | 0.46 | 0.84 | 0.57 | 0.81 | 0.56 |
| Max-Error | 21.91 | 20.11 | 21.45 | 16.93 | 16.07 | 21.2 | 28.61 | 14.18 | 34.65 | 15.05 |
| RMSE | 10.65 | 9.15 | 9.94 | 8.66 | 7.62 | 9.1 | 11.73 | 8.18 | 12.93 | 8.28 |
| BIC | 241.24 | 235.76 | 238.75 | 233.8 | 229.2 | 235.58 | 244.71 | 231.76 | 248.23 | 232.18 |

**Table A5.** Training and test scores on annual basis for the diabetic population.

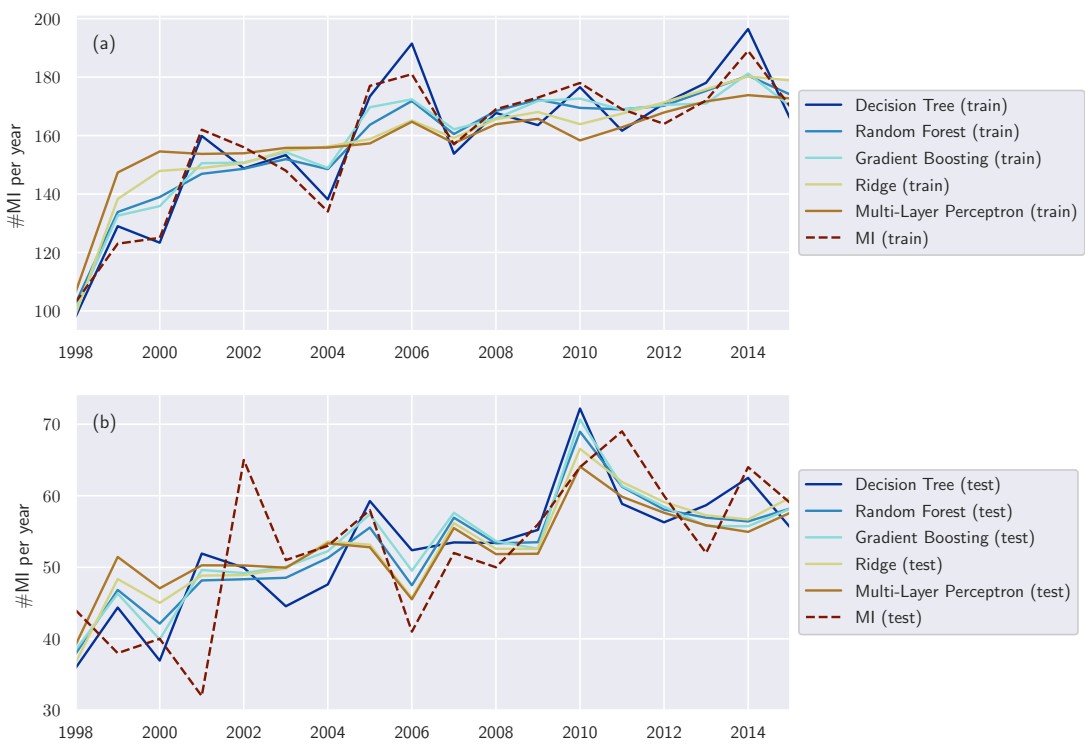

**Figure A5.** Annually aggregated predictions of MI in the smoking population for all models. Predicted (solid) and observed MI (dashed) for training (a) and test (b) sets.

| Method | DTR | | RF | | GBR | | RR | | MLP | |
|---|---|---|---|---|---|---|---|---|---|---|
| Score | Train | Test | Train | Test | Train | Test | Train | Test | Train | Test |
| Adj $R^2$ | 0.93 | 0.33 | 0.86 | 0.48 | 0.9 | 0.45 | 0.72 | 0.46 | 0.61 | 0.39 |
| $R^2$ | 0.93 | 0.35 | 0.86 | 0.49 | 0.9 | 0.47 | 0.72 | 0.48 | 0.62 | 0.41 |
| Max-Error | 10.48 | 19.92 | 15.09 | 16.67 | 14.89 | 17.63 | 22.9 | 16.84 | 29.56 | 18.27 |
| RMSE | 5.79 | 8.15 | 8.52 | 7.18 | 7.2 | 7.38 | 11.94 | 7.27 | 13.95 | 7.73 |
| BIC | 219.32 | 231.62 | 233.2 | 227.05 | 227.14 | 228.02 | 245.37 | 227.49 | 250.95 | 229.69 |

**Table A6.** Training and test scores on annual basis for the smoker population.

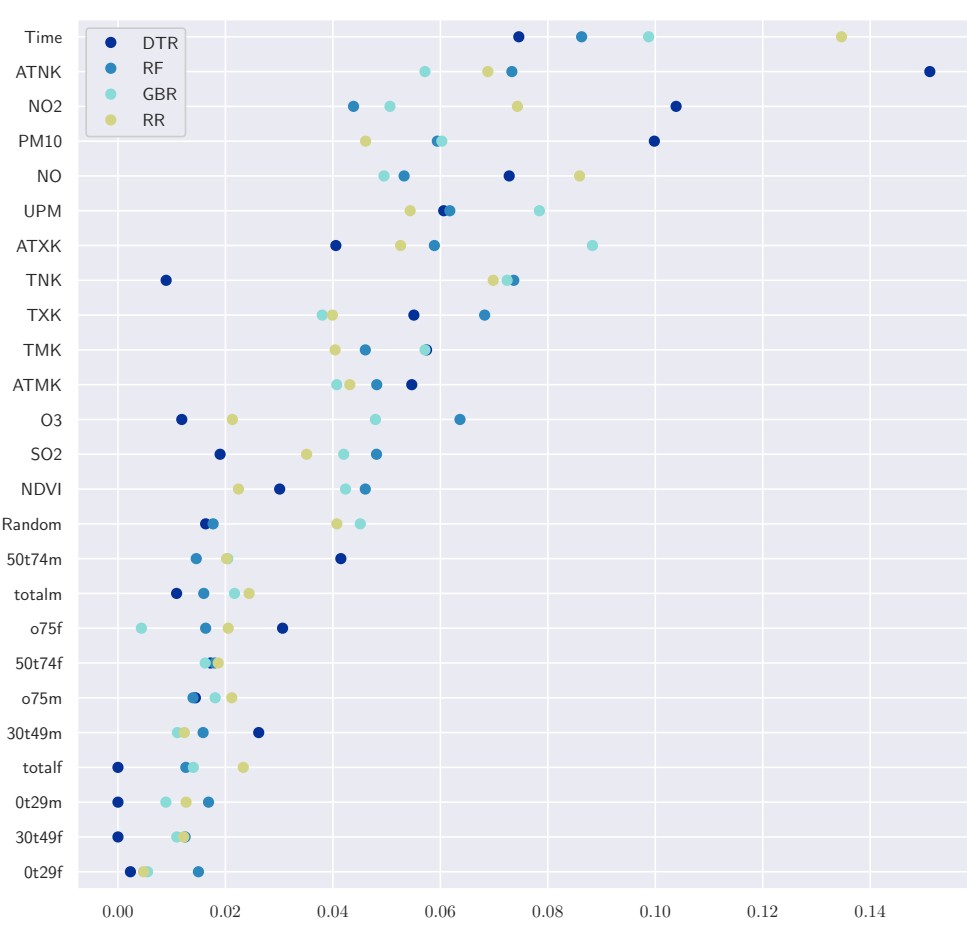

**Figure A6.** Feature importance for predicting MI in the general population. Larger values indicate higher importance and per model the sum over all features equals one.



**Figure A7.** Feature importance for predicting MI in the general population when the full set of demographic features is used. Where applicable the lag in days (0, 1 or 2) is indicated. Larger values indicate higher importance and per model the sum over all features equals one.



**Figure A8.** Feature correlation matrix differentiated by lag in days (0, 1 or 2) and with the full set of demographic features.

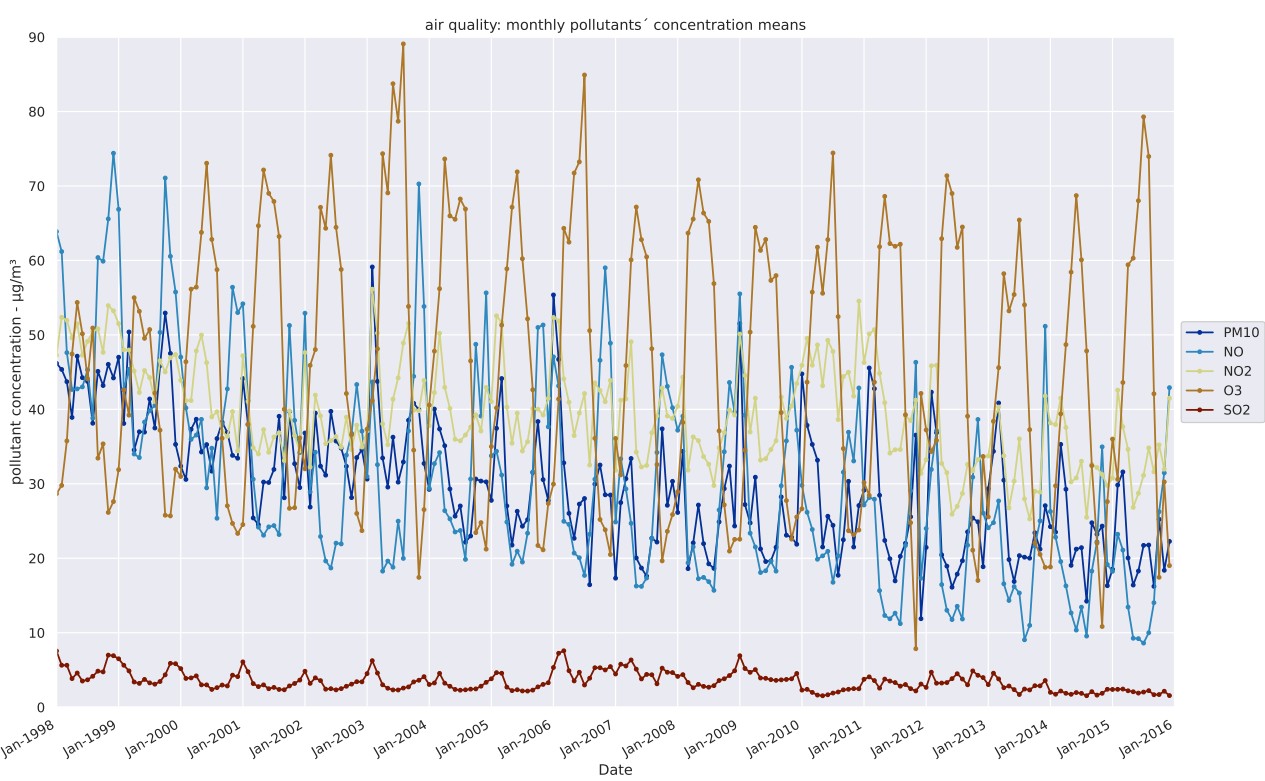

**Figure A9.** Monthly mean time series for all air quality features.



*Code and data availability.* The data used in this paper are available from third party sources. The principle MI registry data is available from the KORA data set, and can be applied for at HMGU here: https://epi.helmholtz-muenchen.de/. Other data sources (environmental and demographic data) are available from the sources quoted in the paper (Table 3). The code used for the ML models and data pre- and

post-processing is available on request from the authors.

*Author contributions.* L.M., W.Z.C., and L.M.B. designed the research, L.M. and M.V. set-up the models and did the simulations, L.M., M.V., and L.M.B. analysed the results and wrote the paper. AS, CM, JL and KW provided the KORA MI data. All authors contributed to the interpretation of results, reviewed the paper, and approved the final draft, and accepted responsibility to submit for publication.

*Competing interests.* The authors have no competing interests.

*Acknowledgements.* We thank our colleague Dr. Gaby Langendijk for her comments that helped to improve this manuscript. This study is part of the Digital Earth project, supported by the Helmholtz Association (funding code ZT-0025).

We would like to thank all members of the Helmholtz Zentrum München, Institute of Epidemiology, and the Chair of Epidemiology at the University Hospital of Augsburg, who were involved in the planning and conduct of the study. Steering partners of the MI Registry, Augsburg, include the Chair of Epidemiology at the University Hospital of Augsburg and the Department of Internal Medicine I, Cardiol-

ogy, University Hospital of Augsburg. Many thanks for their support go to the local health departments, the office-based physicians and the clinicians of the hospitals within the study area. Finally, we express our appreciation to all study participants.

This work was supported by the Helmholtz Zentrum München, German Research Center for Environmental Health, which is funded by the German Federal Ministry of Education, Science, Research and Technology and by the State of Bavaria and the German Federal Ministry

of Health. This research received also supported from the Faculty of Medicine, University of Augsburg, and the University Hospital of Augsburg, Germany. Since the year 2000, the collection of MI data has been co-financed by the German Federal Ministry of Health to provide population-based MI morbidity data for the official German Health Report (see www.gbe-bund.de).



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
