# Peer review of "Machine learning models to predict myocardial infarctions from past climatic and environmental conditions"

_Natural Hazards and Earth System Sciences, 2021_

## Author Response (AR1)

**RC1 - First Review**

Comment 1:

This reviewer is a statistician familiar with the usual epidemiological methods for investigating variation of occurrences of adverse health outcomes over time (usually days) and their application, but with limited familiarity with machine learning methods.

This paper applies machine learning (ML) methods) to obtain a prediction model for MI daily incidence counts from measures of weather and air pollution on the same and three previous days, and demographic data for that year. There is a very substantial literature on dependence on weather and air pollution of variation of occurrences of adverse health outcomes over time (usually days) and on a variety of methods for doing so. However, I have no reason to doubt the authors statement that very few of these have applied ML methods. I had only seen the Zhang 2014 paper cited in this paper, and was eager to see another application.

I learned things from the paper, and was impressed with several aspects of the work. I was unpersuaded that it demonstrated ML was indeed useful in this context, at least from the current analysis and its description, but perhaps that's irrelevant. However, I do believe that my first and most important comment needs addressing before almost any of the results can be usefully.

Response: We thank the reviewer for taking the time to do this thorough review, and for these positive comments, and welcome the suggestions made. We address these one by one, below.

Comment 2:

Most important issue: In all the studies I know of environmental predictors of variation in health outcome over time, control for seasonal and other long term temporal trends is included in the model. This is because these are typically strong predictors, even if demographic changes are allowed for, and otherwise confound the association between environmental variables and outcome. (This includes some publications with several of the same authors as this paper.) This paper appears not to have done so (sorry if I missed it). Assuming not, I could not know how much the importance of the predictors included merely reflected their association with trend or season. For example, given the steep trend in MI counts over the duration of the study (figure 1) I wondered for example if the apparent importance of air pollution might in part be due to a trend in pollution concentration (true of many pollutants in most western European locations).

Perhaps the usual methods for control do not fit easily into the ML framework, but I notice that Zhang (2014) managed by initially discounting the expected counts based on season and year ( "To define the generally expected level of daily mortality counts, we modelled mortality counts as a smooth function (a cubic spline) of day of the year(degrees of freedom =5) while adjusting for day of week and year over the time period of our study (1998–2006)."

Response: We thank the reviewer for this important comment. Controlling for seasonal and long-term trend is indeed a common approach in environmental studies. However, our goal in this study was to adopt a purely data-driven approach, i.e., one where none or very few preconceptions of underlying mechanisms (causal or otherwise) are assumed a priori. Instead, we feed the data as-is into the ML algorithms to see what they can learn from it. Therefore, picking up seasonal signals is essentially part of the overall design of the study.

While the Zhang 2014 paper does correct for such effects by subtracting a baseline for expected mortality, its goal was not to do actual forward projections of the mortality, but instead to identify what predictors are most likely to lead to excess mortality. If forward projections of future mortality were to be carried out on that basis, the subtracted baseline would have to be added again to arrive at the final predictions. The same holds true for our study design. In our case we therefore decided that letting the algorithms decide on how to deal with seasonality, rather than constraining their potential by presupposing any trend/seasonality statistics. Another aspect of this is that our study aims at providing a first step towards projecting MI occurrence under climate change. At such timescales (at least 30 years) seasonal trends may change gradually which would not be reflected by any fixed trend derived from current or historical data. Therefore, extracting the trend from the data alone is a key aspect of our study.

We suggest to address this issue in the revised paper by more clearly outlining this limitation of the study in the introductory section. This includes a more clear and concise description of our reasoning behind choosing data-driven approach and application to the timeseries data.

Comment 3:

The methods description might well be clear to readers well familiar with ML approaches, but if it is designed to be accessible to others some "unpacking" would I think be useful. In particular, the meaning of the "importance" measures of each variable. Zhang 2014 explained this a bit, but even there I was not quite sure. I think it is a measure of reduction in prediction accuracy if the variable in question is dropped from the algorithms' consideration. But if that is right won't the measure be highly dependent on how much remaining variables are associated with the omitted variable? For example apparent and dry-bulb temperature are typically very highly correlated, and absolute humidity pretty highly correlated with both, so dropping one while leaving others in will give a misleading picture of any one variable's importance.

Response: Thank you for the comment. We do agree that a brief but clear description of variable importance is missing from the draft, which we will revise. The notion of variable importance differs depending on context. For some models, such as the linear regression and its variants, these simply relate the magnitude of the trained weights (coefficients) of the model to their associated predictors. In such cases care must be taken to consider the relative magnitudes of the predictors, but this has been addressed in our study by scaling the input data. In other cases, such as the decision tree and variants, the variable importance is based on the mean impurity decrease with respect to splits based on a given

predictor, i.e., by how much the squared sum of deviations from the mean decreases for the observations within the two resulting nodes created by the split. Additionally, there are other measures that work by systematically dropping features and observing the decrease in prediction accuracy. In this study however, we did not use these, but rather relied on the built-in importances of the models as described above.

We suggest to address this in the paper by outlining the variable importance used for each of the regression methods applied.

Comment 4:

No descriptive data is given for the outcome data. It would be usual and is I believe useful to give measures like daily mean, SD, min and max counts overall and for each region.

Response: We agree that such a table would give the reader the opportunity to more easily get an overview of the results at first glance. We will add the proposed information in a table in the appendix of the paper.

Comment 5:

Readers would be further informed if the comparisons of prediction accuracy could include a conventional (a priori selected predictors) time series regression model. For example that used by Chen (Eur H J 2019, with some of the same authors as the current paper). Also, for the comparisons of annual predictions, it would be very useful to state how much prediction accuracy did ALL the environmental variables add to that with just the demographic variables.

Response: We thank the reviewer for bringing up these two important issues. We believe that conducting a thorough time series analysis based on distributed lag non-linear models is out of the scope of the current paper. While we believe that a direct comparison of the methods on the same data would be very insightful, we see it as a potential step for further research. Instead we would like to address this first issue by adding a comparison of the methods to the paper, outlining differences as well as similarities of both approaches.

For the second issue, we propose to run an additional experiment with all demographic predictors turned off and to compare in the revised paper the prediction accuracy to the original results. In the interest of speed and brevity we would propose to do this only for the overall population.

Comment 6:

I found the paper much longer than I thought necessary to make it's main points, and believe its length will put off many readers.

Response: We also agree that the draft could be made more concise. We propose to shorten it substantially, without taking away the main points of the paper.

Comment 7:

I am aware of a couple of reviews of the temporal association of MI with weather and pollution, which could I think be usefully cited. Sun Env Poll 2018 and Mustafie JAMA 2012.

Response: We thank the reviewer for making these two good suggestions. We propose to reference these in the appropriate sections of the paper.

**RC2 - Second Review**

Comment 1:

This paper evaluate different machine algorithm to predict myocardial infarctions (MI) using environmental exposures variables. The paperi is well structured and clear. The authors found a poor performance on predicting daily or weekly MI levels, but a good perfoemne on predicting the yearly values. I think the analysis and the intrperetation of the results could be improved considering the following aspects:}

Response: We thank the reviewer for the review and these positive comments. We address the comments made, below.

Comment 2:

The authors considered a linear respresentation of the exposure (features). This is highly questionable especially for metereological variables. It is well known that the effect of ambient temperature on several health outcomes is non linear with higher health impact for coldest and hottest temperature. This could be modelled using splines reparametrisation or in a simpler way using 3-piece linear spline (segmented linear) function (e.g. see Armstrong, Ben Models for the Relationship Between Ambient Temperature and Daily Mortality, Epidemiology: November 2006 - Volume 17 - Issue 6 - p 624-631). Mi suggestioni s to consider a reparametrisation of tem perature related variables considring both the heat and cold effects.

Response: We thank the reviewer for bringing up this important issue. Unlike more traditional methods in epidemiology we do indeed not make any prior assumption on the form of the exposure-response relationship between variables (meteorological or otherwise). This is part of the data-driven study design, where little to no preconceptions of underlying mechanisms are assumed beforehand. Moreover, of the methods we have applied in this paper only ridge regression is a linear method. Decision Trees, Random Forests, Gradient Boosting as well as Multi-Layer Perceptron can tackle highly-nonlinear problems. The exposure-response models are very well suited to time series modelling as displayed in the Armstrong paper. Instead of a time series modelling approach, we use an approach based on multivariate machine learning regression models. These models do not require the presupposition of a known exposure-response relationship. We also point out that this study is aimed towards developing models to make long-term tendency projections at climate-timescales (i.e., 30 years). At such timescales underlying statistical properties may

change gradually which would not be reflected by any prescribed exposure-response function derived from historical or current data. Instead, we hope that by letting the models pick these up based on the provided data alone, the application to ensemble data of climate simulations will provide an improved generalization. We also believe that it is of legitimate scientific interest to apply and evaluate models other than those that have been traditionally used, especially in the light of the success ML models have seen in other branches of science and technology.

We propose to address this in the revised paper by clarifying the reasoning behind our study design and giving a more concise overview of the differences to time series modelling, clearly pointing out limitations such as this one.

Comment 3:

The authors considered a lag of 3 days, and this would be enough for the heat effect of temperature and for most pollutants, but it has been shown that the cold effect could have a longer delayed effects (up to 3 or 4 weeks). I suggest to increase the lag up to at least 21 days.}

Response: We thank the reviewer for this excellent suggestion. However, increasing lag to 21 days for all variables would greatly increase the computational effort required to run the simulations and especially to conduct the hyperparameter tuning. We therefore propose to address this issue by increasing lag of only the temperature variables to 21 days and repeating the experiments, by adding another predictor, that estimates cold exposure during the past 21 days (e.g., overall minimum temperature during those 21 days).

Comment 4:

I think that the good performance at yearly level is totally expcted as you are considering year and day of the year as features in your models, so environmental features doesn't seems having any role here. To see if the environmental features have a role on predicting the yearly values models without time variables should be tested.

Response: Here, the reviewer is raising an important issue. Adding the year as a feature understandably casts doubt on the relevance of environmental and other predictors in this study design, suggesting the model may simply learn the desired answers based on temporal predictors, effectively acting as a lookup-table. Before proposing a solution to this weakness, we would like to point out two factors here. First, the analysis of the variable importance does not support the notion that predictors other than year and day of the year are irrelevant. In fact, environmental predictors come out on top of the relevant predictors across all experiments carried out. Specifically, the year is shown in Figure A7 to have only low to medium relevance.

Second, we have taken great care to make sure that the models do not use any of the test (validation) data during training, i.e., to prevent data leakage. The correct responses could therefore not just be memorized by the models, because they have never seen them during the training step.

However, to accommodate this comment we propose to remove the year as a predictor entirely, repeating all experiments. While this comes with substantial effort, we believe that rectifying this issue would increase the robustness of the study results, and we will describe any differences.

Comment 5:

Some demographic features were considered and I think they are reatevely stable over time, so I think they shouldn't contain any information at daily or weekly level, Changes on the demograhic structure should be captured by the trend (year) variable. If the objective was to standardise the outcome the authors could consider to use as outcome the daily incidence of MI, perhaps considering logarithm values in order to have a more symmetric distribution, without considering demoraphic predictors.

Response: The reviewer is correct that demographic predictors generally do not undergo large changes in a matter of days or weeks. However, as explained in the methods section our models by construction require daily input for all predictors. This is a technical limitation that we can not change or alleviate. We expect changing to annual values for all days in the year would not change the ML predictions, and moreover, we likely are closer to the 'true' value of population and other demographic changes as they happen gradually within the year in the exposed population.

Comment 6:

I diagree with sentences in line 90-95. Actually it is possible considering case-controls desing nested within time-series using case-crossover design, that is each case is matched with days before and after the case day and the association measured conditioning on those risk set.

Response: We thank the reviewer for this important correction. We will change the sentence accordingly.

---

## Author Response (AR2)

**Paper NHESS 2021-389**

"Machine learning models to predict myocardial infarctions from past climatic and environmental conditions"

Response to Reviewer #1:

We thank the reviewer for the additional review, and these comments. We did on purpose not include "year" as a variable, as can also be seen from Table 3. As time indicators, we only included week, day, and month. Any (inter)annual variability that is present in the data, would be resulting from the environmental and demographic data, which is also precisely the purpose of the paper: a data-driven approach to estimate MI occurrence based on those data (and not time). Note also that from the analysis of variable importance, these time indicators are not as relevant as most environmental and demographic variables (see Figure 5).

We acknowledge that external effects not considered by the model may indirectly be picked up and be incorrectly attributed to the environmental and demographic predictors instead. If applied to projections of future climate change this may lead to over- oder underestimation of these effects by the models.

We therefore propose to add the following clarification to the paper (around Line 423 on Page 19):

"Further analysis of the data, including accounting for trends over time, may further increase robustness of the results to prevent the attribution of exogenous effects not considered in the model to the existing features."

We hope the proposed additional sentence is a good solution to this issue raised by the reviewer.

Response to Reviewer #2:

We thank the reviewer for the additional review, and the encouraging comment that we have completed the revision in good order.